



# Impact of bias nonstationarity on the performance of uni- and multivariate bias-adjusting methods

Jorn Van de Velde[1,2], Matthias Demuzere[3,1], Bernard De Baets[2], and Niko E. C. Verhoest[1]

[1]Hydro-Climatic Extremes Lab, Ghent University, Ghent, Belgium
[2]KERMIT, Department of Data Analysis and Mathematical Modelling, Ghent University, Ghent, Belgium
[3]Department of Geography, Ruhr-University Bochum, Bochum, Germany

**Correspondence:** Jorn Van de Velde (jorn.vandevelde@ugent.be)

**Abstract.**

Climate change is one of the biggest challenges currently faced by society, with an impact on many systems, such as the hydrological cycle. To locally assess this impact, Regional Climate Model (RCM) simulations are often used as input for hydrological rainfall-runoff models. However, RCM results are still biased with respect to the observations. Many methods have been developed to adjust these biases, but only during the last few years, methods to adjust biases that account for the correlation between the variables have been proposed. This correlation adjustment is especially important for compound event impact analysis. As a simple example of those compound events, hydrological impact assessment is used here, as hydrological models often need multiple locally unbiased input variables to ensure an unbiased output. However, it has been suggested that multivariate bias-adjusting methods may perform poorly under climate change conditions because of bias nonstationarity. In this study, two univariate and three multivariate bias-adjusting methods are compared with respect to their performance under climate change conditions. To this end, the methods are calibrated in the late 20th century (1970-1989) and validated in the early 21st century (1998-2017), in which the effect of climate change is already visible. The variables adjusted are precipitation, evaporation and temperature, of which the former two are used as input for a rainfall-runoff model, to allow for the validation of the methods on discharge. Although not used for discharge modelling, temperature is a commonly-adjusted variable in both uni- and multivariate settings and therefore important to take into account. The methods are also evaluated using indices based on the adjusted variables, the temporal structure, and the multivariate correlation. For precipitation, all methods decrease the bias in a comparable manner. However, for many other indices the results differ considerably between the bias-adjusting methods. The multivariate methods often perform worse than the univariate methods, a result that is especially notable for temperature and evaporation. As these variables have already changed the most under climate change conditions, this reinforces the opinion that the multivariate bias-adjusting methods are not yet fit to cope with nonstationary climate conditions. Although the effect is slightly dampened by the hydrological model, our analysis still reveals that, to date, the simpler univariate bias-adjusting methods are preferred for assessing climate change impact.





## 1  Introduction

The influence of climate change is felt throughout many regions of the world, as becomes evident from the higher frequency
or intensity of natural hazards, such as floods, droughts, heatwaves and forest fires (IPCC, 2012). As these intensified natural
hazards threaten society, it is essential to be prepared for them. Knowledge on future climate change is obtained by running
Global Climate Models (GCMs), creating large ensemble outputs such as in the Climate Model Intercomparison Project 6
(CMIP6) (Eyring et al., 2016). Although they are informative on a global scale, the generated data are too coarse for local

climate change impact assessments. To bridge the gap from the global to the local scale, Regional Climate Models have
become a standard application (Jacob et al., 2014), using the output from GCMs as input or boundary conditions.

   Although the information provided by both GCMs and RCMs is very valuable, both are biased with respect to the observa-
tions, especially for precipitation (Kotlarski et al., 2014). The biases can occur in any statistic and are commonly defined as *"a
systematic difference between a simulated climate statistic and the corresponding real-world climate statistic"* (Maraun, 2016).

These biases are caused by temporal or spatial discretisation and unresolved or unrepresented physical processes (Teutschbein
and Seibert, 2012; Cannon, 2016). An important example of the latter is convective precipitation, which can only be resolved by
very high resolution models. Although the improvement of models is an important area of research (Prein et al., 2015; Kendon
et al., 2017; Helsen et al., 2019; Fosser et al., 2020), such improved models are computationally expensive. As such, it is still
necessary practice to statistically adapt the climate model output to adjust the biases (Christensen et al., 2008; Teutschbein and

Seibert, 2012; Maraun, 2016).

   Many different bias-adjusting methods exist (Teutschbein and Seibert, 2012; Gutiérrez et al., 2019). They all calibrate a
transfer function using the historical simulations and historical observations and apply this transfer function to the future
simulations to generate future 'observed values' or an adjusted future. Of all the different methods, the quantile mapping
method (Panofsky et al., 1958) was shown to be the generally best performing method (Rojas et al., 2011; Gudmundsson et al.,

2012). Quantile mapping adjusts biases in the full distribution, whereas most other methods only adjust biases in the mean
and/or variance.

   An important problem with quantile mapping and most other commonly used methods is that they are univariate and do
not adjust biases in the multivariate correlation. Although quantile mapping can retain climate model multivariate correlation
(Wilcke et al., 2013), the ability of univariate methods to improve the climate model's multivariate correlation has been ques-

tioned (Hagemann et al., 2011; Ehret et al., 2012; Hewitson et al., 2014). This is important for impact assessment, as local
impact models often need multiple input variables and many high-impact events are caused by the co-occurrence of multiple
phenomena, the so-called 'compound events' (Zscheischler et al., 2018, 2020). For example, floods can be characterised by a
rainfall-runoff model using evaporation and precipitation time series as an input. If the correlation between these variables is
biased with respect to the observations, then it can be expected that the model output is biased as well. This results in a higher

uncertainty when using these models and thus in the resulting assessment. During the past decade, multiple methods have been
developed to counter this problem. The first methods focused on the adjustment of two jointly occurring variables, most often
precipitation and temperature, such as those by Piani and Haerter (2012) and Li et al. (2014). However, it became clear that ad-





justing only two variables would not suffice, hence many more methods have been developed that jointly adjust more variables, including those by Vrac and Friederichs (2015); Cannon (2016); Mehrotra and Sharma (2016); Dekens et al. (2017); Cannon (2018); Vrac (2018); Nguyen et al. (2018); Robin et al. (2019). Yet, the recent growth in availability of such methods comes along with a gap in the knowledge on their performance. In some studies, these methods have been compared with one or two older multivariate methods to reveal the improvements (Vrac and Friederichs, 2015; Cannon, 2018) or with univariate methods (Räty et al., 2018; Zscheischler et al., 2019; Meyer et al., 2019). Each of these three studies indicates that the univariate and multivariate methods lead to different results, yet it is difficult to conclude whether uni- or multivariate methods perform best. According to Zscheischler et al. (2019) multivariate methods have an added value. Räty et al. (2018) conclude that the multivariate methods and univariate methods performed similarly, while Meyer et al. (2019) could not draw definitive conclusions. These studies vary in set-up, adjusted variables and study area, which all could have caused the difference in added value. In all three studies, the same method, namely the Multivariate Bias Correction in $n$ dimensions (MBCn) (Cannon, 2018) was the basis for comparison. Only recently, the first studies comparing multiple multivariate bias-adjusting methods were published (François et al., 2020; Guo et al., 2020). The study by François et al. (2020) focused on the different principles underlying the multivariate bias-adjusting methods and concluded that the choice of method should be based on the end user's goal. Besides, they also noticed that so far, all multivariate methods fail in representing the temporal structure of a time series. In contrast to the focus of François et al. (2020), Guo et al. (2020) studied the performance of multivariate bias-adjusting methods for climate change impact assessment and concluded that multivariate methods could be interesting in this context. However, they also noticed that the performance of the multivariate methods was lower in the more recent validation period and suggested that this could be caused by bias nonstationarity. As the use of multivariate bias-adjusting methods could be an important tool for climate change impact assessment, this deserves more attention.

The bias stationarity - or bias time invariance - assumption is the most important assumption for bias correction. It implies that the bias is the same in the calibration and validation or future periods and that the transfer function based on the calibration period can consequently be used in the future period. However, this assumption does not hold due to different types of nonstationarity induced by climate change, which may cause problems (Milly et al., 2008; Derbyshire, 2017). In the context of bias adjustment, this problem has been known for several years (Christensen et al., 2008; Ehret et al., 2012), but has not received a lot of attention. A few authors have tried to propose new types of bias relationships (Buser et al., 2009; Ho et al., 2012; Sunyer et al., 2014; Kerkhoff et al., 2014). Recently, it has been suggested that it is best to assume a non-monotonic bias change (Van Schaeybroeck and Vannitsem, 2016). Some authors suggested that bias nonstationarity could be an important source of uncertainty (Chen et al., 2015; Velázquez et al., 2015; Wang et al., 2018; Hui et al., 2019), but not all found clear indications of bias nonstationarity (Maraun, 2012; Piani et al., 2010; Maurer et al., 2013).

The availability of new methods and more data enables a more coherent assessment of the bias (non)stationarity issue. By comparing three bias-adjusting methods in a climate change context with possible bias nonstationarity, some of the remaining questions in François et al. (2020) and Guo et al. (2020) can be answered. The three multivariate bias-adjusting methods that will be compared in this study are 'Multivariate Recursive Quantile Nesting Bias Correction' (MRQNBC, Mehrotra and Sharma (2016)), MBCn (Cannon, 2018) and 'dynamical Optimal Transport Correction' (dOTC, Robin et al. (2019)). These three





methods give a broad view of the different multivariate bias adjustment principles, which we will elaborate on in Section 3.3.
As a baseline, two univariate bias-adjusting methods will be used: Quantile Delta Mapping (QDM, Cannon et al. (2015)) and
modified Quantile Delta Mapping (mQDM, Pham (2016). QDM is a classical univariate bias-adjusting method and is chosen
for this analysis as it is a robust and relatively common quantile mapping method, especially as one of the subroutines in the
multivariate bias-adjusting methods (Mehrotra and Sharma, 2016; Nguyen et al., 2016; Cannon, 2018). mQDM, on the other
hand, is one of the so-called 'delta change' methods, which are based on an adjustment of the historical time series. Using
these univariate bias-adjusting methods, we can assess whether multivariate and univariate bias-adjusting methods differ in
their response to possible bias nonstationarity.

The methods will be compared by applying them for the bias adjustment of precipitation, potential evaporation and temperature. The bias-adjusted time series will be used as inputs for a hydrological model in order to simulate the discharge. Discharge
time series are the basis for flood hazard calculation, but can also be considered as an interesting source of validation themselves (Hakala et al., 2018). Although temperature is not needed as an input for the hydrological model, it is, together with
precipitation, the most common variable to be adjusted in similar studies and therefore it is also included here. In order to
mimic climate change context, the 'historical' or calibration time series runs from 1970 to 1989 and the 'future' or validation
time series runs from 1998 to 2017, which is only recent past. In the latter time frame, effects of climate change are already
visible (IPCC, 2013). The change of some biases from calibration to validation time series will be calculated, to indicate the
extent of the bias nonstationarity. Maurer et al. (2013) proposed the R index for this purpose (see Section 2.4). Calculating the
bias nonstationarity between both periods will give an indication of the impact of a changing bias on climate impact studies
for the end of the 21st century. As Chen et al. (2015) mentioned: *"If biases are not constant over two very close time periods,
there is little hope they will be stationary for periods separated by 50 to 100 years"*

## 2 Data and validation

### 2.1 Data

The observational data used were obtained from the Belgian Royal Meteorological Institute (RMI) Uccle observatory. The
most important time series used is the 10-min precipitation amount, gauged with a Hellmann-Fuess pluviograph, from 1898 to
2018. An earlier version of this precipitation dataset was described by Demarée (2003) and analyzed in De Jongh et al. (2006).
Multiple other studies have used this time series (Verhoest et al., 1997; Verstraeten et al., 2006; Vandenberghe et al., 2011;
Willems, 2013). The 10-min precipitation time series was aggregated to daily level to be comparable with the other time series
used.

For the multivariate method, the precipitation time series was combined with a 2 meter air temperature and potential evaporation time series. The daily potential evaporation was calculated by the RMI from 1901 to 2019, using the Penman formula
for a grass reference surface (Penman, 1948) with variables measured at the Uccle observatory. Daily average temperatures
were obtained using measurements from 1901 to 2019. As the last complete year for precipitation was 2017, the data were
used from 1901 to 2017, amounting to 117 years of daily data.




The IPCC report (IPCC, 2013) clearly states the influence of climate change on different variables. For Belgium, this is illustrated by Fig. 1, in which the temperature and evaporation anomalies for the 21st century are all higher than the long-term mean value. However, for precipitation, the effect of climate change is not yet visible.

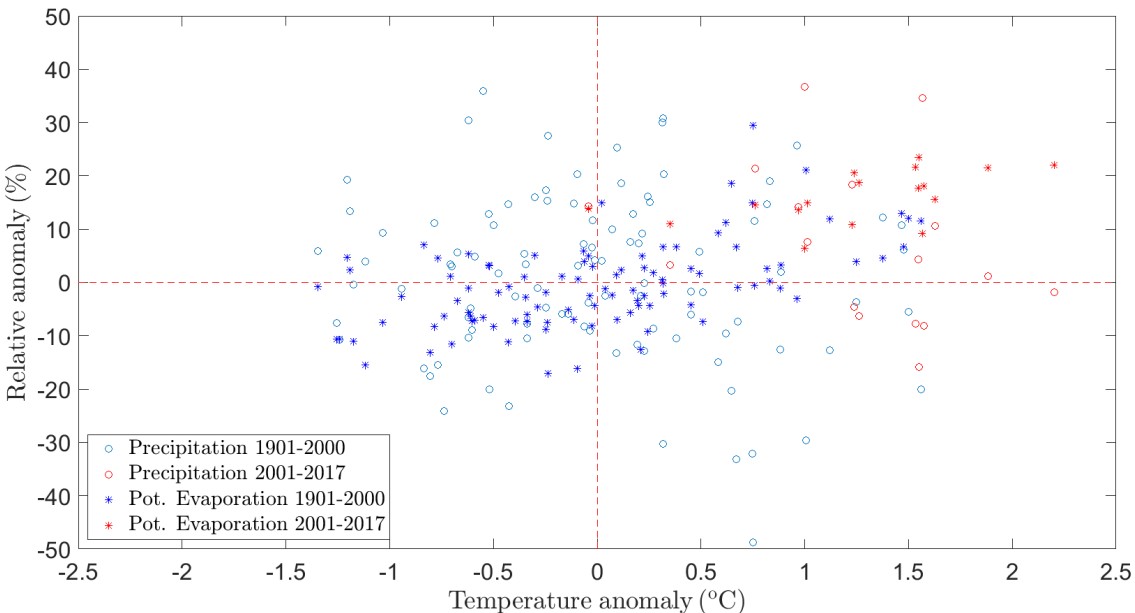

**Figure 1.** Yearly mean temperature, precipitation and evaporation anomalies for 1901-2017, compared with long-term mean value from 1920-1980. Red points are 21st century values.

For the simulations, data from the EURO-CORDEX project (Jacob et al., 2014) were used. The Rossby Centre regional
climate model RCA4 was used (Strandberg et al., 2015) as it is one of the few RCMs with potential evaporation as an output variable. This RCM was forced with boundary conditions from the MPI-ESM-LR GCM (Popke et al., 2013). Historical data and scenario data for the grid cell comprising Uccle were respectively obtained for 1970-2005 and 2006-2100. The former time frame is limited by the earliest available data from the RCM. The latter time frame was only used until 2017, in accordance with the observational data. As climate change scenario, an RCP4.5 forcing was used in this paper (Van Vuuren et al., 2011).
Since only 'near future' (from the model point of view) data were used, the choice of forcing does not have a large impact. However, when studying scenarios in a time frame further away from the present, using an ensemble of forcings is more relevant to be aware of the uncertainty regarding future climate change impact. Evaluations of the RCA4 model have shown that there is a bias in precipitation, especially in winter (Strandberg et al., 2015), but this bias is in line with the biases from other EURO-CORDEX models (Kotlarski et al., 2014).
As Uccle (near Brussels) is situated in a region with small topographic differences, it is assumed that the conditions in Uccle can be applied anywhere within the climate model grid cell and the variance within this cell is about the same. This assumption





can be made as long as the resulting adjusted data are not used for extremely localized studies, such as urban hydrology impact assessment.

## 2.2 Time frames

As mentioned in the introduction, it is important to assess bias-adjusting methods in a context they will be used in, i.e. under climate change conditions. The time series used in this study were chosen accordingly: 1970-1989 was chosen as the 'historical' or calibration time period and 1998-2017 was chosen as the 'future' or validation time period. Time series of 20 years were chosen here, although it is advised to use 30 years of data to have robust calculations (Berg et al., 2012; Reiter et al., 2018). However, as no climate model data prior to 1970 are available, using 30 years of data would have led to overlapping time series.

## 2.3 Validation framework

An important aspect in bias adjustment is the validation of the methods. Different methods are available, of which a pseudo-reality experiment (Maraun, 2012) is one of the most-used ones. In this method, each member of a model ensemble is in turn used as the reference in a cross-validation. However, while such a set-up is useful when comparing bias-adjustment methods, it only mimics a real application context. When sufficient observations are available, a 'pseudo-projection' setup

(Li et al., 2010) can be used. This set-up resembles a 'differential split-sample testing' (Klemeš, 1986) and is more in agreement with a practical application of bias-adjusting methods. Differential split-sample testing has been used in a bias adjustment context by Teutschbein and Seibert (2013), by constructing two time series with respectively the driest and wettest years. In our case study, it is assumed that the two time series differ enough because of climate change. Consequently, the approach is simple, and as the validation is not set in the future, it is considered a 'pseudo-projection'.

Besides the choice of time frames and data, also the choice of validation indices is of key importance. Maraun and Widmann (2018a) stress that these indices should only be indirectly affected by the bias adjustment, as only validating on adjusted indices can be misleading. Such adjusted indices are the precipitation intensity, temperature and evaporation, which are used to build the transfer function in the historical setting and should be corrected by construction. Under bias stationarity, this correction will be carried over to the future, possibly hiding small inconsistencies that may arise for extreme values. If the bias is not

stationary, the effect might be different between adjusted and indirectly affected indices. As such, besides the three adjusted variables (indices 1 to 3 in Table 1) and their correlations (indices 4 to 12, which are directly adjusted by some of the methods), also indices based on the precipitation occurrence and on the discharge $Q$ are used. The occurrence-based indices (13 to 16) allow for assessing how the methods influence the precipitation time series structure. The discharge-based indices (17 and 18) allow for the assessment of the impact of the different bias-adjusting methods on simulated river flow. The discharge-based

indices combine the information of the other indices by routing through the rainfall-runoff model. They are the most important aspect of the assessment, as they indicate the natural hazard. ETCCDI (Expert Team on Climate Change Detection and Indices) precipitation indices (Zhang et al., 2011) have also been considered and calculated. However, these are not included in this paper, as the differences in ETCCDI indices were minor and did not allow to clearly discern between the different methods. All





indices were calculated taking all days into account, instead of only calculating them on wet days, as some of the multivariate
bias-adjusting methods do not discriminate between wet or dry days in their adjustment.

**Table 1.** Overview of the indices used

| Nr | Index | Name |
|----|-------|------|
| 1 | $P_x$ | Precipitation amount percentile values, with $x$ the percentile considered |
| 2 | $T_x$ | Temperature percentile values, with $x$ the percentile considered |
| 3 | $E_x$ | Evaporation percentile values, with $x$ the percentile considered |
| 4 | $\mathrm{corr}_{P,E}$ | Spearman correlation between the time series of $P$ and $E$ |
| 5 | $\mathrm{corr}_{P,T}$ | Spearman correlation between the time series of $P$ and $T$ |
| 6 | $\mathrm{corr}_{E,T}$ | Spearman correlation between the time series of $E$ and $T$ |
| 7 | $\mathrm{crosscorr}_{P,E,0}$ | Lag-0 crosscorrelation between the time series of $P$ and $E$ |
| 8 | $\mathrm{crosscorr}_{P,T,0}$ | Lag-0 crosscorrelation between the time series of $P$ and $T$ |
| 9 | $\mathrm{crosscorr}_{E,T,0}$ | Lag-0 crosscorrelation between the time series of $E$ and $T$ |
| 10 | $\mathrm{crosscorr}_{P,E,1}$ | Lag-1 crosscorrelation between the time series of $P$ and $E$ |
| 11 | $\mathrm{crosscorr}_{P,T,1}$ | Lag-1 crosscorrelation between the time series of $P$ and $T$ |
| 12 | $\mathrm{crosscorr}_{E,T,1}$ | Lag-1 crosscorrelation between the time series of $E$ and $T$ |
| 13 | $P_{P00}$ | Precipitation transition probability from a dry to a dry day |
| 14 | $P_{P10}$ | Precipitation transition probability from a wet to a dry day |
| 15 | $N_{\mathrm{dry}}$ | Number of dry days |
| 16 | $P_{\mathrm{lag1}}$ | Precipitation lag-1 auto-correlation |
| 17 | $Q_x$ | Discharge percentiles, with $x$ the percentile considered |
| 18 | $Q_{T20}$ | 20-year return period value of discharge |

## 2.4 Bias nonstationarity

In a study on possible changes in bias, Maurer et al. (2013) proposed the R index:

$$\mathrm{R} = 2\frac{\mid \mathrm{bias_f} - \mathrm{bias_h} \mid}{\mid \mathrm{bias_f} \mid + \mid \mathrm{bias_h} \mid}, \tag{1}$$

where $\mathrm{bias_f}$ and $\mathrm{bias_h}$ are the biases in respectively the future and historical time series, calculated on the basis of the observations and raw climate simulations. The R index takes a value between 0 and 2. If the index is greater than one, the difference in bias between the two sets is larger than the average bias of the model and it is likely that the bias adjustment would degrade the RCM output rather than improve it. The index is calculated for the indices used for validation in order to have an indication of the influence of bias nonstationarity on these indices. Besides for the indices, the R index is also calculated for the average and standard deviation of each variable, in order to be able to more easily visualise the changes in distribution.





## 2.5 Hydrological model

Similar to Pham et al. (2018), we use the Probability Distributed Model (PDM, Moore (2007)), a lumped conceptual rainfall-runoff model to calculate the discharge for the Grote Nete watershed in Belgium. This model uses precipitation and evaporation time series as inputs to generate a discharge time series. The PDM as used here was calibrated by Cabus (2008) using the Particle Swarm Optimization algorithm (PSO, Eberhart and Kennedy (1995)). As in Pham et al. (2018), it was assumed that the differences between meteorological conditions in the Grote Nete-watershed and Uccle are negligible, and that thus the adjusted data for the Uccle grid cell can be used as a forcing for the PDM. Furthermore, the goal is not to make predictions, but to assess the impact of different bias adjustment methods on the discharge values. To calculate the bias on the indices, observed, raw and adjusted RCM time series were used as forcing for this model. The discharge time series generated by the observations is considered to be the 'observed' discharge, and biases are calculated in comparison with this time series.

## 2.6 Validation metrics

The residual biases relative to the observations and to the model bias are often used in this paper to graphically present and interpret the results. These residual biases are based on the 'added value' concept (Di Luca et al., 2015) and enable a comparison based on two aspects. The first aspect is the performance in removing the bias, the second is the extent of the bias removal in comparison with the original value for the corresponding index for the observation time series. The use of the residual biases allows for a detailed study and comparison of the effect of bias adjustment on the different indices.

The residual bias relative to the observations $\mathrm{RB_O}$ for an index $k$ is calculated as follows:

$$\mathrm{RB_O}\,(k) = 1 - \frac{|\,\mathrm{bias}_{\mathrm{raw}(k)}\,| - |\,\mathrm{bias}_{\mathrm{adj}(k)}\,|}{|\,\mathrm{obs}\,(k)\,|}, \tag{2}$$

with raw($k$) the raw climate model simulations, adj($k$) the adjusted climate model simulations and obs($k$) the observed values for index $k$.

The residual bias relative to the model bias $\mathrm{RB_{MB}}$ for an index $k$ is calculated as follows:

$$\mathrm{RB_{MB}}\,(k) = 1 - \frac{|\,\mathrm{bias}_{\mathrm{raw}(k)}\,| - |\,\mathrm{bias}_{\mathrm{adj}(k)}\,|}{|\,\mathrm{bias}_{\mathrm{raw}(k)}\,|}. \tag{3}$$

Absolute values are used in Eqs. (2) and (3) to compute the absolute difference between the raw and adjusted values, thus neglecting a possible change of sign of the bias. If the values of these residual biases are lower than 1 for an index, the method performs better than the raw RCM for this index. The best methods have low scores on both residual biases for as many indices as possible.

## 3 Bias-adjusting methods

### 3.1 Occurrence-bias adjustment: Thresholding

One of the deficiencies of RCMs, especially in Northwest Europe, are the so-called 'drizzle days' (Gutowski et al., 2003; Themeßl et al., 2012; Argüeso et al., 2013), i.e. the simulation of a small amount of precipitation on days that are supposed





to be dry. This has an influence on the temporal structure of the simulated time series and should thus be adjusted (Ines and Hansen, 2006). This is commonly done in an occurrence-bias-adjusting step before the main step, the intensity-bias adjustment. In this study, we use the threshlding occurrence-bias-adjusting method. Thresholding is one of the most common occurrence-bias-adjusting methods and has been in use for many years (e.g. Hay and Clark (2003); Schmidli et al. (2006); Ines and Hansen (2006)). This method is only applicable in regions where the assumption holds that the simulated time series has more wet
days than the observed time series. This is the case for Northwest Europe (Themeßl et al., 2012) and Belgium in particular. An advanced version of the thresholding method is used here. To adjust the number of wet days, the frequencies of dry days in the observations and in the simulations are calculated. The difference between the two frequencies, $\Delta N$, is the number of days of the simulated time series that have to be adapted. The simulated series is adapted by first sorting the wet days and thus only changing the $\Delta N$ lowest days of the simulation time series by setting them to 0. $\Delta N$ is computed for the past and applied in
the future and consequently relies on the bias stationarity assumption. However, as thresholding is used prior to all methods, the influence of possible bias nonstationarity on $\Delta N$ is assumed to be negligible.

In this advanced version of thresholding, some considerations are made. First, a day is considered wet if its simulated precipitation amount is above 0.1 mm, to account for measurement errors in the observations. Second, the adjustment is done on a monthly basis, to withhold a realistic temporal structure. This implies that to correct the number of wet days in month
$m$, all days of month $m$ of the time series are selected. Third, both historical and future simulations are adjusted during the same calculation step, to ensure a sound comparison during the intensity phase of the adjustment. If only either the historical or future time series would have been adjusted, the assumption that the bias can be transferred from the historical to the future time period would be impaired. The thresholding method is summarized in Algorithm 1.





---

**Algorithm 1** Thresholding

**Input:**

    Historical observations $X^{\mathrm{ho}}$

    Historical simulations $X^{\mathrm{hs}}$

    Future simulations $X^{\mathrm{fs}}$

**Output:**

    Adjusted historical $X^{\mathrm{hs}}_{\mathrm{out}}$ and future simulations $X^{\mathrm{fs}}_{\mathrm{out}}$

    Initialization

    **for** $m = 1:12$ **do**

        Select data for month $m$: $X^{\mathrm{ho}}_m$, $X^{\mathrm{hs}}_m$ and $X^{\mathrm{fs}}_m$ {Loop over months}

        Calculate the percentage of dry days in month $m$ for the historical observations

        Calculate the percentage of dry days in month $m$ for the historical simulations

        Calculate $\Delta N$ for month $m$

        {Adjustment of historical simulated time series}

        Select and sort the wet days

        Set the $\Delta N$ wet days with the lowest precipitation amount to 0

        Restore the original order of the wet days of month $m$

        Restore the full historical time series for month $m$

        {Adjustment of future simulated time series}

        Select and sort the wet days

        Set the $\Delta N$ wet days with the lowest precipitation amount to 0

        Restore the original order of the wet days of month $m$

        Restore the full future time series for month $m$

        {Reconstruction}

        Replace the data in $X^{\mathrm{hs}}_{\mathrm{out}}$ with the adjusted data for month $m$

        Replace the data in $X^{\mathrm{fs}}_{\mathrm{out}}$ with the adjusted data for month $m$

    **end for**

---

## 3.2 Univariate intensity-bias-adjusting methods

### 3.2.1 Quantile Delta Mapping

The Quantile Delta Mapping (QDM) method was first proposed by Li et al. (2010). Its main idea is to preserve the climate simulation trends: it takes trend nonstationarity (changes in the simulated distribution) into account to a certain degree. Although it handles temperature adjustments well, it gives unrealistic values for precipitation and was therefore extended by Wang and



Chen (2014) for precipitation adjustment. A comparison with other quantile mapping methods by Cannon et al. (2015) showed

this method to perform best with respect to the preservation of trends. Cannon et al. (2015) bundled both the method by Li et al. (2010) (*Equidistant CDF-matching*) and Wang and Chen (2014) (*Equiratio CDF-matching*) under the name *Quantile Delta Mapping*, because of the similarity with delta change methods (which are described in e.g. Olsson et al. (2009), Willems and Vrac (2011) and Räty et al. (2014)).

Mathematically, this method can be written as

$$x_i^{\mathrm{fa}} = x_i^{\mathrm{fs}} + F_{x^{\mathrm{ho}}}^{-1}\left(F_{x^{\mathrm{fs}}}\left(x^{\mathrm{fs}}\right)\right) - F_{x^{\mathrm{hs}}}^{-1}\left(F_{x^{\mathrm{fs}}}\left(x^{\mathrm{fs}}\right)\right) \tag{4}$$

in the additive case, and

$$x_i^{\mathrm{fa}} = x_i^{\mathrm{fs}} \frac{F_{x^{\mathrm{ho}}}^{-1}\left(F_{x^{\mathrm{fs}}}\left(x^{\mathrm{fs}}\right)\right)}{F_{x^{\mathrm{hs}}}^{-1}\left(F_{x^{\mathrm{fs}}}\left(x^{\mathrm{fs}}\right)\right)} \tag{5}$$

in the ratio or multiplicative case. The superscripts hs, ho, fs and fa indicate respectively the historical simulations, the historical observations, the future simulations and the adjusted future. In this paper, the additive version is used for tem-

perature time series and the multiplicative one for precipitation and evaporation time series. This choice is based on the work of Wang and Chen (2014), who have shown that using the additive adjustment for precipitation results in unrealistic precipitation values and introduced a multiplicative adjustment. For evaporation, we follow the few available studies (e.g. Lenderink et al. (2007)) in using the same adjustment as for precipitation.

To ensure the consistency of the time series, Themeßl et al. (2011) implemented a 61-day moving window. Here, a 91-day

moving window is opted for, as suggested by Rajczak et al. (2016) and Reiter et al. (2018). This enables the adjustment of each day based on $91\,\mathrm{days/year} \cdot 20\,\mathrm{years} = 1820\,\mathrm{days}$. These days were used to build an empirical CDF (as in Gudmundsson et al. (2012); Gutjahr and Heinemann (2013), among others), because of the ease of application. The number of quantiles implemented was determined automatically by the Matlab function `ecdf`, ranging between 200 and 400. It is also important to note that for precipitation, Eq. (5) was applied only on the days considered wet, i.e. with a precipitation higher than 0.1 mm.

For consistency, a threshold of 0.1 mm was also used for evaporation. As no prior examples for evaporation adjustment were available, we assumed this consistency was the best option. It is important to note that although QDM is only applied on wet days, it can still transform low-precipitation wet days into days considered dry (e.g. with a precipitation amount < 0.1 mm) if the ratio in Eq. (5) is small enough.

### 3.2.2 Modified Quantile Delta Mapping

Pham (2016) proposed another version of QDM, following the delta change philosophy (Olsson et al., 2009; Willems and Vrac, 2011): the trend established by the RCM is assumed to be more thrust-worthy than the absolute value itself. When applying this type of methods, the simulated change between the historical and the future is applied to the observations. Thus, instead of the future simulations, the historical observations are adjusted to the future 'observations'. As Johnson and Sharma (2011) mention, this workflow could be problematic for future impact assessment, as it inherits the temporal structure of the historical

observations. This method is mathematically very similar to the QDM method, exchanging the roles of $x^{\mathrm{fs}}$ and $x^{\mathrm{ho}}$. Thus, it is





named 'modified Quantile Delta Mapping' (mQDM), and can for the additive case be written as

$$x_i^{\text{fa}} = x_i^{\text{ho}} + F_{x^{\text{fs}}}^{-1}\left(F_{x^{\text{ho}}}\left(x^{\text{ho}}\right)\right) - F_{x^{\text{hs}}}^{-1}\left(F_{x^{\text{ho}}}\left(x^{\text{ho}}\right)\right). \tag{6}$$

The ratio version is mathematically written as

$$x_i^{\text{fa}} = x_i^{\text{ho}} \frac{F_{x^{\text{fs}}}^{-1}\left(F_{x^{\text{ho}}}\left(x^{\text{ho}}\right)\right)}{F_{x^{\text{hs}}}^{-1}\left(F_{x^{\text{ho}}}\left(x^{\text{ho}}\right)\right)}. \tag{7}$$

For the implementation, the same principles were used as for the QDM method: a 91-day moving window, empirical CDFs and a threshold of 0.1 mm/day to be considered as a wet day.

### 3.3   Multivariate intensity-bias-adjusting methods

The increasing number of multivariate bias-adjusting methods throughout the 2010s urges the need to classify them according to their properties. One possible classification was done by Vrac (2018), who proposed the 'marginal/dependence' versus the
'successive conditional' approach. The former approach separately adjusts the 1D-marginal distributions and the dependence structure and is applied in e.g. Vrac and Friederichs (2015), Cannon (2018) and Vrac (2018). These two components are then recombined to obtain data that are close to the observations for both marginal and multivariate aspects. The latter approach consists of adjusting one given variable and then adjusting a second variable conditionally on the second variable: this procedure is applied successively to each variable. Examples can be found in e.g. Piani and Haerter (2012), Li et al. (2014) and
Dekens et al. (2017). According to Vrac (2018), the latter approach suffers from two main limitations. First, the adjustment is performed conditionally on the previously adjusted data. However, the adjustment is often applied in bins. As a result, for each variable, the amount of data available for each bin decreases, thus decreasing the robustness of the adjustment. Second, the ordering of the variables in the successive adjustments matters. For example, Li et al. (2014) point out that their 'Joint Bias Correction for temperature' (JBCt) and 'Joint Bias Correction for precipitation' (JBCp) methods, which respectively first ad-
just temperature and precipitation, differ in performance. For these two reasons, Vrac (2018) advocates for the use of the more robust and coherent 'marginal/dependence' approach. Robin et al. (2019) and François et al. (2020) extended this classification by introducing the all-in-one approach, which adjusts the marginal variables and the correlations simultaneously, 'dynamical Optimal Transport Correction' (dOTC) (Robin et al., 2019) being such a method.

Another perspective on the multivariate bias-adjusting methods is to consider the amount of temporal adjustment that is
allowed or applied by the method. This is important, as the amount of temporal adjustment is intrinsically linked with the main goal, the adjustment of the multivariate distribution of the variables. This distribution, in which the dependence is characterised by the underlying copula (Nelsen, 2006; Schölzel and Friederichs, 2008), can be estimated using the ranks. Thus, to adjust the multivariate distribution, the ranks of the climate model are replaced by those of the observations, using methods such as the 'Schaake Shuffle' (Clark et al., 2004; Vrac and Friederichs, 2015). This implies that the temporal structure and trends of
the climate model will be altered, which may have a considerable impact (François et al., 2020). This impact is especially large when multiday characteristics strongly matter, such as in applications as the hydrological example we use in this study (Addor and Seibert, 2014). Vrac (2018) mentions this necessity to modify the temporal structure and rank chronology of the





simulations. Yet, he also mentions that the extent of this modification is still a matter of debate. Cannon (2016) describes this as
the 'knobs' that control whether marginal distributions, inter-variable or spatial dependence structure and temporal structure are
more informed by the climate model or the observations. Thus, the choice between the temporal structure of the climate model
and unbiased dependence structures is a trade-off that has to be made. Some methods, such as those by Vrac and Friederichs
(2015), Mehrotra and Sharma (2016) and Nguyen et al. (2018) rely on the observations for their temporal properties, while
other methods try to find the middle ground (e.g. Vrac (2018) and Cannon (2018)).

    Our choice of multivariate bias-adjusting methods takes the above classification into account. The oldest method in the com-
parison is 'Multivariate Recursive Quantile Nesting Bias Correction' (Mehrotra and Sharma, 2016). This method completely
replaces the simulated correlations by those of the observations and is a 'marginal/dependence' method according to François
et al. (2020). 'Multivariate Bias Correction in $n$ dimensions' (Cannon, 2018) is both a 'marginal/dependence' method and a
method that tries to combine information from the climate model and the observations. The most recent method, 'dynamical
Optimal Transport Correction' (Robin et al., 2019) differs considerably from the other two methods: it generalises the 'transfer
function'-principle using the 'optimal transport' paradigm (Villani, 2008), thereby defining a new category of multivariate bias-
adjusting methods: the above-mentioned all-in-one approach. Although far from complete, by comparing these three methods,
a broad view of the different approaches in multivariate bias adjustment can be obtained.

### 3.3.1   Multivariate Recursive Quantile Nesting Bias Correction

In 2016, Mehrotra and Sharma proposed a new multivariate bias adjustment method, named 'Multivariate Recursive Quantile
Nesting Bias Correction' (MRQNBC), based on a combination of several older methods by Johnson and Sharma (2012),
Mehrotra and Sharma (2012) and Mehrotra and Sharma (2015). The underlying idea of this method is to adjust on more than
one timescale, an idea that most bias-adjusting methods do not incorporate (Haerter et al., 2011). This adjustment on multiple
timescales is applied by adjusting the biases in lag-0- and lag-1-auto and the cross-correlation coefficients, i.e. the persistence
attributes, instead of focusing on the mean or the distribution.

As a first step in this method, QDM is applied separately on variables to adjust the empirical CDFs. This is followed by a
multivariate bias adjustment to adjust the lag-0 and lag-1 auto and cross-correlation coefficients. This combination of univariate
and multivariate bias-adjusting methods is applied on all time scales. For the multivariate adjustment, two models are used: a
multivariate first-order autoregressive (AR(1)) model with constant parameters at the daily and yearly level, and a multivariate
AR(1) model with periodic parameters (Salas, 1980) at the monthly and seasonal level. All steps are applied to the different
types of data: historical observations of temperature, evaporation and precipitation (combined in the matrix $\mathbf{X}^{\mathrm{ho}}$), historical
climate model simulations of the three variables (the matrix $\mathbf{X}^{\mathrm{hs}}$) and climate model projections of the three variables (the
matrix $\mathbf{X}^{\mathrm{fs}}$), which have to be adjusted. All these datasets are of size $T \times N$, with $T$ the number of time steps and $N$ the
number of variables.

    The quantile-mapped future GCM time series for time step $t$ is denoted as $\mathbf{X}^{\mathrm{fa}}_t$. The standardised versions of this time series
and of the observed time series are denoted as $\check{\mathbf{X}}^{\mathrm{fa}}_t$ and $\check{\mathbf{X}}^{\mathrm{ho}}_t$, respectively. Using the standardised time (zero mean and unit
variance) series, the multivariate AR(1) model with constant parameters (MAR) for the observed and GCM multivariate time





series can be expressed as (Mehrotra and Sharma, 2016):

$$\breve{\mathbf{X}}_t^{\text{ho}} = \mathbf{C}\breve{\mathbf{X}}_{t-1}^{\text{ho}} + \mathbf{D}\epsilon_t \tag{8}$$

and

$$\breve{\mathbf{X}}_t^{\text{fa}} = \mathbf{E}\breve{\mathbf{X}}_{t-1}^{\text{fa}} + \mathbf{F}\epsilon_t, \tag{9}$$

with $\mathbf{C}$ and $\mathbf{D}$ the coefficient matrices of $\breve{\mathbf{X}}_t^{\text{ho}}$, $\mathbf{E}$ and $\mathbf{F}$ the coefficient matrices of $\breve{\mathbf{X}}_t^{\text{fa}}$ and $\epsilon_t$ a white noise term. The coefficient matrices are calculated using the $N \times N$ lag-0 and lag-1 cross-correlation matrices $\mathbf{M}_0$ and $\mathbf{M}_1$. Using the standardised time series, the elements of these matrices can be expressed as (Salas, 1980):

$$m_0^{i,j} = \frac{1}{T}\sum_{t=1}^{T} x_t^i x_t^j \tag{10a}$$

$$m_1^{i,j} = \frac{1}{T-1}\sum_{t=1}^{T} x_t^i x_{t+1}^j, \tag{10b}$$

with $i$ and $j$ the column numbers of $\breve{\mathbf{X}}_t$, referring to the variables whose correlation is calculated. This enables the calculation of $\mathbf{C}$ and $\mathbf{E}$ as (Matalas, 1967):

$$\mathbf{C} = \mathbf{M}_1^{\text{ho}}\mathbf{M}_0^{\text{ho}-1}, \tag{11a}$$

$$\mathbf{E} = \mathbf{M}_1^{\text{fa}}\mathbf{M}_0^{\text{fa}-1}, \tag{11b}$$

and of $\mathbf{D}$ and $\mathbf{F}$ via

$$\mathbf{D}\mathbf{D}^{\text{T}} = \mathbf{M}_0^{\text{ho}} - \mathbf{M}_1^{\text{ho}}\mathbf{M}_0^{\text{ho}-1}\mathbf{M}_1^{\text{ho}\,\text{T}} \tag{12a}$$

$$\mathbf{F}\mathbf{F}^{\text{T}} = \mathbf{M}_0^{\text{fa}} - \mathbf{M}_1^{\text{fa}}\mathbf{M}_0^{\text{fa}-1}\mathbf{M}_1^{\text{fa}\,\text{T}}, \tag{12b}$$

which can be solved using the eigenvalues and eigenvectors of $\mathbf{D}\mathbf{D}^{\text{T}}$ or $\mathbf{F}\mathbf{F}^{\text{T}}$:

$$\mathbf{D} = \mathbf{V}\sqrt{\mathbf{S}}\mathbf{V}^{\text{T}}, \tag{13a}$$

$$\mathbf{F} = \mathbf{V}\sqrt{\mathbf{S}}\mathbf{V}^{\text{T}}, \tag{13b}$$

with $\mathbf{V}$ the matrix of eigenvectors and $\mathbf{S}$ a diagonal matrix with the corresponding eigenvalues.

The multivariate bias adjustment is then implemented by removing the lag-0 and lag-1 auto- and cross-correlations from the future time series $\breve{\mathbf{X}}_t^{\text{fa}}$ (the matrices $\mathbf{E}$ and $\mathbf{F}$) and applying the observed lag-0 and lag-1 auto- and cross-correlations ($\mathbf{C}$ and $\mathbf{D}$) to the future time series and thus creating a modified future time series, $\breve{\mathbf{X}}_t'^{\text{fa}}$. These steps are applied by first rearranging and simplifying Eq. (9) for $\epsilon_t$:

$$\epsilon_t = \mathbf{F}^{-1}\left(\breve{\mathbf{X}}_t^{\text{fa}} - \mathbf{E}\breve{\mathbf{X}}_{t-1}^{\text{fa}}\right), \tag{14}$$





with $\epsilon_t$ now a standardised vector of $N$ variables calculated by removing the lag-0 and lag-1 auto- and cross-correlations from the $\breve{\mathbf{X}}_t^{\mathrm{fa}}$ time series. This vector is plugged into Eq. (8) along with the matrices $\mathbf{C}$ and $\mathbf{D}$ in which $\breve{\mathbf{X}}_t^{\mathrm{fa}}$ is used instead of $\breve{\mathbf{X}}_t^{\mathrm{ho}}$ to obtain the modified time series:

$$\breve{\mathbf{X}}_t'^{\mathrm{fa}} = \mathbf{C}\breve{\mathbf{X}}_{t-1}'^{\mathrm{fa}} + \mathbf{D}\mathbf{F}^{-1}\left(\breve{\mathbf{X}}_t^{\mathrm{fa}} - \mathbf{E}\breve{\mathbf{X}}_{t-1}^{\mathrm{fa}}\right), \tag{15}$$

which can be rearranged as:

$$\breve{\mathbf{X}}_t'^{\mathrm{fa}} = \mathbf{C}\breve{\mathbf{X}}_{t-1}'^{\mathrm{fa}} + \mathbf{D}\mathbf{F}^{-1}\breve{\mathbf{X}}_t^{\mathrm{fa}} - \mathbf{D}\mathbf{F}^{-1}\mathbf{E}\breve{\mathbf{X}}_{t-1}^{\mathrm{fa}}. \tag{16}$$

This model preserves the observed persistence attributes. As a last step, destandardising results in the bias-adjusted time series $\mathbf{X}_t'^{\mathrm{fa}}$.

When using the multivariate AR(1) with periodic parameters (PMAR), the parameters are derived separately for each period to allow for periodicity. In this case, the vectors $\mathbf{X}_{t,\tau}^{\mathrm{ho}}$ and $\mathbf{X}_{t,\tau}^{\mathrm{fa}}$ respectively represent the observed and quantile mapped GCM time series. The subscript $t$ refers to the year and the subscript $\tau$ to a specific period in the year.

The elements of the periodic version of $\mathbf{M}_0$ and $\mathbf{M}_1$ can be calculated as (Salas, 1980):

$$m_{0,\tau}^{i,j} = \frac{\sum_{t=1}^{T_\tau}\left(x_{t,\tau}^i - \bar{x}_\tau^i\right)\left(x_{t,\tau}^j - \bar{x}_\tau^j\right)}{T_\tau s_\tau^i s_\tau^j} \tag{17a}$$

$$m_{1,\tau}^{i,j} = \frac{\sum_{t=1}^{T_\tau}\left(x_{t,\tau}^i - \bar{x}_\tau^i\right)\left(x_{t,\tau-1}^j - \bar{x}_{\tau-1}^j\right)}{T_\tau s_\tau^i s_{\tau-1}^j}, \tag{17b}$$

with $T_\tau$ the number of time steps of the period $\tau$, $\bar{x}_\tau$ and $\bar{x}_{\tau-1}$ the mean of periods $\tau$ and $\tau-1$ (for instance, if $\tau$ is summer, than $\tau-1$ is spring) and $s_\tau$ and $s_{\tau-1}$ the standard deviations of periods $\tau$ and $\tau-1$. The correlation matrices are calculated in the same way as in the non-periodic steps. The only difference is that they are calculated for every period (e.g. separately for every season or month). For every time step in period $\tau$, the corresponding value can be adjusted as follows to preserve the observed persistence attributes:

$$\breve{\mathbf{X}}_{t,\tau}'^{\mathrm{fa}} = \mathbf{C}_\tau\breve{\mathbf{X}}_{t,\tau-1}'^{\mathrm{fa}} + \mathbf{D}_\tau\mathbf{F}_\tau^{-1}\breve{\mathbf{X}}_{t,\tau}^{\mathrm{fa}} - \mathbf{D}_\tau\mathbf{F}_\tau^{-1}\mathbf{E}_\tau\breve{\mathbf{X}}_{t,\tau-1}^{\mathrm{fa}}. \tag{18}$$

The different time steps are combined with the nesting method proposed in Johnson and Sharma (2012) and Mehrotra and Sharma (2015). First, QDM (as described in Section 3.2.1) is applied at a daily level, followed by MAR. These adjusted time series are then aggregated and averaged to form a monthly time series, which is adjusted by QDM, standardised and adjusted by PMAR. Note that the standardisation of the aggregated time series does not imply that the variables of a period $\tau$ of that time series have zero mean and unit variance. The results of the monthly adjustment are aggregated and averaged to form seasonal time series, which are also adjusted using QDM, standardised and adjusted by PMAR. As a last nesting step, the results are once more aggregated and averaged to build an annual time series, which is adjusted using QDM and MAR. The outcomes of





all these steps are combined into a weighting factor that is used to modify the daily time series accordingly (Srikanthan and
Pegram, 2009):

$$\mathbf{X}''^{\text{fa}}_{t,j,s,i} = \left(\frac{\mathbf{Y}'^{\text{fa}}_{j,s,i}}{\mathbf{Y}^{\text{fa}}_{j,s,i}}\right)\left(\frac{\mathbf{Z}'^{\text{fa}}_{s,i}}{\mathbf{Z}^{\text{fa}}_{s,i}}\right)\left(\frac{\mathbf{A}'^{\text{fa}}_{i}}{\mathbf{A}^{\text{fa}}_{i}}\right)\mathbf{X}'^{\text{fa}}_{t,j,s,i}, \tag{19}$$

with $t$ the day, $j$ the month, $s$ the season, $i$ the year, $\mathbf{Y}'^{\text{fa}}_{j,s,i}$ the monthly adjusted value, $\mathbf{Y}^{\text{fa}}_{j,s,i}$ the aggregated-averaged monthly
value, $\mathbf{Z}'^{\text{fa}}_{s,i}$ the seasonal adjusted value, $\mathbf{Z}^{\text{fa}}_{s,i}$ the aggregated-averaged seasonal value, $\mathbf{A}'^{\text{fa}}_{i}$ the adjusted yearly value and $\mathbf{A}^{\text{fa}}_{i}$
the aggregated-averaged yearly value. The full procedure is summarised in Algorithm 2.

---

**Algorithm 2** MRQNBC

**Input:**

    Daily historical observations $\mathbf{X}^{\text{ho}}$

    Daily historical simulations $\mathbf{X}^{\text{hs}}$

    Daily future simulations $\mathbf{X}^{\text{fs}}$

**Output:**

    Adjusted future simulations $\mathbf{X}''^{\text{fa}}$

    **for** #Timescales **do**

        Apply QDM to calculate $\mathbf{X}^{\text{fa}}$

        Standardise $\mathbf{X}^{\text{ho}}$ and $\mathbf{X}^{\text{fa}}$

        Calculate matrices $\mathbf{C}$ and $\mathbf{D}$ of $\check{\mathbf{X}}^{\text{ho}}$ and $\mathbf{E}$ and $\mathbf{F}$ of $\check{\mathbf{X}}^{\text{fa}}$ {The calculations of $\mathbf{C}$, $\mathbf{D}$, $\mathbf{E}$ and $\mathbf{F}$ depend on the periodicity
        of the timescale}

        Apply the persistence adjustment to calculate $\check{\mathbf{X}}'^{\text{fa}}$

        Destandardise $\check{\mathbf{X}}'^{\text{fa}}$

        Aggregate $\mathbf{X}^{\text{ho}}$ and $\mathbf{X}'^{\text{fa}}$ to the higher timescale {Except for the yearly timescale}

    **end for**

    Calculate weighting factors for all timescales except the daily timescale

    Calculate the final adjusted daily value $\mathbf{X}''^{\text{fa}}$

---

The nesting method cannot fully remove biases at all time scales, thus Mehrotra and Sharma (2016) suggested to repeat the
complete procedure multiple times. However, in our case this seemed to exacerbate the results, so the method was run only
once.



### 3.3.2 Multivariate Bias Correction in $n$ dimensions

In 2018, Cannon (2018) proposed the 'Multivariate Bias correction in $n$ dimensions' (MBCn) method as a flexible multivariate
bias-adjusting method. The method's flexibility has attracted some attention, as it has already been used in multiple studies
(Räty et al., 2018; Zscheischler et al., 2019; Meyer et al., 2019; François et al., 2020). This method consists of three steps.
First, the multivariate data are rotated using a randomly generated orthogonal rotation matrix, adjusted with the additive form
of QDM, and rotated back until the calibration period model simulations converge to the observations. This convergence is
verified on the basis of the energy distance (Rizzo and Székely, 2016). Second, the validation period simulations are adjusted
using QDM, as this method preserves the simulated trends. As the last step, these adjusted time series are shuffled using the
Schaake Shuffle (Clark et al., 2004) based on the rank order of the rotated dataset.

Considering the $j$-th iteration of the method, denoted by the subscript $[j]$, the first step consists of rotating the data sets
using an $N \times N$ randomly generated orthogonal rotation matrix $\mathbf{R}_{[j]}$. This orthogonal rotation matrix was created using the
algorithm by Mezzadri (2007, pg. 597). This rotation is formulated as

$$
\begin{aligned}
\tilde{\mathbf{X}}^{\mathrm{hs}}_{[j]} &= \mathbf{X}^{\mathrm{hs}}_{[j]} \mathbf{R}_{[j]} \\
\tilde{\mathbf{X}}^{\mathrm{fs}}_{[j]} &= \mathbf{X}^{\mathrm{fs}}_{[j]} \mathbf{R}_{[j]} \\
\tilde{\mathbf{X}}^{\mathrm{ho}}_{[j]} &= \mathbf{X}^{\mathrm{ho}}_{[j]} \mathbf{R}_{[j]}
\end{aligned}
\tag{20}
$$

with $\tilde{\mathbf{X}}_{[j]}$ the resulting rotated matrix. In the next step, additive quantile delta mapping is applied to each variable in $\tilde{\mathbf{X}}^{\mathrm{hs}}_{[j]}$ and
$\tilde{\mathbf{X}}^{\mathrm{fs}}_{[j]}$, using the corresponding variable in $\tilde{\mathbf{X}}^{\mathrm{ho}}_{[j]}$ as the target. The resulting matrices $\mathbf{X}^{\mathrm{hc}}_{[j]}$ and $\mathbf{X}^{\mathrm{fa}}_{[j]}$ are rotated back:

$$
\begin{aligned}
\mathbf{X}^{\mathrm{hs}}_{[j+1]} &= \mathbf{X}^{\mathrm{hc}}_{[j]} \mathbf{R}^{-1}_{[j]} \\
\mathbf{X}^{\mathrm{fs}}_{[j+1]} &= \mathbf{X}^{\mathrm{fa}}_{[j]} \mathbf{R}^{-1}_{[j]} \\
\mathbf{X}^{\mathrm{ho}}_{[j+1]} &= \mathbf{X}^{\mathrm{ho}}_{[j]}
\end{aligned}
\tag{21}
$$

These steps have to be repeated until the multivariate distribution of $\mathbf{X}^{\mathrm{hs}}_{[j+1]}$ matches $\mathbf{X}^{\mathrm{ho}}$. The similarity is measured using
the (squared) energy distance (Székely and Rizzo, 2004, 2013; Rizzo and Székely, 2016), a measure of statistical discrepancy
between two multivariate distributions. For two $N$-dimensional independent random vectors $\mathbf{x}$ and $\mathbf{y}$ with respective CDFs $F$
and $G$, this measure is given by:

$$
D^2(F,G) = 2E\left[\| \mathbf{x} - \mathbf{y} \|\right] - E\left[\| \mathbf{x} - \mathbf{x}' \|\right] - E\left[\| \mathbf{y} - \mathbf{y}' \|\right] \geq 0
\tag{22}
$$





with $E$ the expected value, $|| \, . \, ||$ the Euclidean norm and $\mathbf{x}'$ and $\mathbf{x}$ and $\mathbf{y}'$ and $\mathbf{y}$ i.i.d.. A practical way of calculating this measure is as follows (Székely and Rizzo, 2013), with $\mathbf{X} = (X_1, ..., X_{n_1})$ and $\mathbf{Y} = (Y_1, ... Y_{n_2})$:

$$
\begin{aligned}
D\left(\mathbf{X}, \mathbf{Y}\right) = & \frac{2}{n_1 n_2} \sum_{i=1}^{n_1} \sum_{m=1}^{n_2} \mid X_i - Y_m \mid \\
& - \frac{1}{n_1^2} \sum_{i=1}^{n_1} \sum_{j=1}^{n_1} \mid X_i - X_j \mid \\
& - \frac{1}{n_2^2} \sum_{l=1}^{n_2} \sum_{m=1}^{n_2} \mid Y_l - Y_m \mid ,
\end{aligned}
\tag{23}
$$

with $i$, $j$, $l$ and $m$ denoting the time steps.

As a last step, the preservation of trends of QDM has to be combined with the restoration of the multivariate ranks by the transformations. To do this, first either the additive or multiplicative version of quantile delta mapping (depending on the variable) has to be applied to each variable of the original data set $\mathbf{X}^{\text{fs}}$, using $\mathbf{X}^{\text{hs}}$ and $\mathbf{X}^{\text{ho}}$ as historical baseline data. As a final step, the elements of each column of $\mathbf{X}^{\text{fa}}$ are reordered following a method known as the 'Schaake Shuffle' (Clark et al., 2004; Vrac and Friederichs, 2015). In this method, the ranks of each variable of $\mathbf{X}^{\text{fa}}$ are swapped with the ranks of the corresponding variables of $\mathbf{X}^{\text{fs}}_{[j+1]}$, thus reordering the time series' structure according to the ranks of the observations. The Schaake Shuffle can be mathematically formulated as follows (Clark et al., 2004). Let $\mathbf{X}$ be a vector of $n$ time steps of a variable, and $\boldsymbol{\chi}$ be the sorted vector of $\mathbf{X}$, that is:

$$
\mathbf{X} = (x_1, x_2, ... x_n), \text{ and} \tag{24}
$$

$$
\boldsymbol{\chi} = \left(x_{(1)}, x_{(2)}, ... x_{(n)}\right), \quad x_{(1)} \leq x_{(2)} ... \leq x_{(n)} \tag{25}
$$

Let $\mathbf{Y}$ be a vector of $n$ historical observations and $\boldsymbol{\gamma}$ be the sorted vector of $\mathbf{Y}$:

$$
\mathbf{Y} = (y_1, y_2, ... y_n), \text{ and} \tag{26}
$$

$$
\boldsymbol{\gamma} = \left(y_{(1)}, y_{(2)}, ... y_{(n)}\right), \quad y_{(1)} \leq y_{(2)} ... \leq y_{(n)}. \tag{27}
$$

$\mathbf{B}$ is then the vector of indices describing the original observation number that corresponds to the values in the ordered vector $\boldsymbol{\gamma}$. The main step of the Schaake Shuffle is to reconstruct the reordered vector $\mathbf{X}^{\text{ss}}$:

$$
\mathbf{X}^{\text{ss}} = (x_1^{\text{ss}}, x_2^{\text{ss}}, ..., x_n^{\text{ss}}) \tag{28}
$$

where

$$
x_q^{\text{ss}} = x_{(r)}, \, q = \mathbf{B}\left[r\right], \text{and } r = 1, ..., n. \tag{29}
$$

In MBCn, $\mathbf{X}^{\text{fa}}$ is sorted according to $\mathbf{X}^{\text{fs}}_{[j+1]}$ instead of respectively $\mathbf{X}$ and $\mathbf{Y}$, resulting in a final adjusted data set $\mathbf{X}^{\text{fa}\prime}$. To account for ties in this procedure, a small random value is added before calculating the ranks of $\mathbf{X}^{\text{fs}}_{[j+1]}$. The version of QDM





applied in the step prior to the Schaake Shuffle is the same as in Section 3.2.1. As such, the shuffling procedure is the only difference with the univariate bias-adjustment, implying that differences in performance can be related to it.

The MBCn method was shown by Cannon (2018) to outperform many earlier multivariate bias-adjusting methods, such as the EC-BC (Vrac and Friederichs, 2015), the JBC (Li et al., 2014), MBCr and MBCp methods (Cannon, 2016). In contrast, Cannon (2018) also pointed out some problems. Depending on the number of variables, the computational cost can get too high and convergence speed too low, or overfitting might become an issue. The first problem can be tackled by implementing sufficient time steps when having a lot of variables. Second, to address the convergence speed, Pitié et al. (2007) suggested

using a deterministic selection of rotation matrices that maximizes the distance between rotation axis sets instead of randomly generating them. It is also suggested by Cannon (2018) to use the most efficient form of quantile mapping and to limit the use of an advanced quantile mapping method to the last step. Third, to avoid overfitting (and to reduce the computational cost), early stopping is also suggested by Cannon (2018) (e.g. Prechelt (1998)). As the number of variables is limited in this case, overfitting did not seem to be a problem. Yet, to reduce unnecessary computational costs, the similarity in consecutive

energy distances was used as a measure to stop the computation. A tolerance of 0.0001 was used: if the difference between two consecutively calculated energy distances was lower, the computation was halted. The full procedure is summarised in Algorithm 3.

---

**Algorithm 3** MBCn

---

**Input:**

    Historical observations $\mathbf{X}^{\mathrm{ho}}$

    Historical simulations $\mathbf{X}^{\mathrm{hs}}$

    Future simulations $\mathbf{X}^{\mathrm{fs}}$

**Output:**

    Adjusted future simulations $\mathbf{X}'^{\mathrm{fa}}$

 

    Initialisation: tolerance $\epsilon$ and initial energy distance difference $\Delta D_0$

    **while** $\Delta D > \epsilon$ **do**

        Randomly generate a rotation matrix $\mathbf{R}$

        Rotate $\mathbf{X}^{\mathrm{ho}}$, $\mathbf{X}^{\mathrm{hs}}$ and $\mathbf{X}^{\mathrm{fs}}$

        Apply the additive form of QDM

        Rotate $\mathbf{X}^{\mathrm{ho}}$, $\mathbf{X}^{\mathrm{hs}}$ and $\mathbf{X}^{\mathrm{fs}}$ back

        Calculate the energy distance $D$ between $\mathbf{X}^{\mathrm{hs}}$ and $\mathbf{X}^{\mathrm{ho}}$

        Calculate the decrease in energy distance $\Delta D$

    **end while**

    Apply QDM to the original inputs to calculate $\mathbf{X}^{\mathrm{fa}}$

    Apply the Schaake Shuffle based on the rotated future simulations to calculate $\mathbf{X}'^{\mathrm{fa}}$

---





### 3.3.3 Dynamical Optimal Transport Correction

Recently, Robin et al. (2019) indicated that the notion of a transfer function in quantile mapping can be generalised to the
theory of optimal transport. Optimal transport is a way to measure the dissimilarity between two probability distributions and
to use this as a means for transforming the distributions in the most optimal way (Villani, 2008; Peyré and Cuturi, 2019).

Optimal transport was used by Robin et al. (2019) to adjust the bias of a multivariate data set in the 'dynamical Optimal
Transport Correction' method (dOTC), which extends the 'CDF-transform' (CDF-t) bias-adjusting method (Michelangeli et al.,
2009). dOTC calculates the optimal transport plans from $\mathbf{X}^{\mathrm{ho}}$ to $\mathbf{X}^{\mathrm{hs}}$ (the bias between the model and the simulations) and
from $\mathbf{X}^{\mathrm{hs}}$ to $\mathbf{X}^{\mathrm{fs}}$ (the evolution of the model). The combination of both optimal transport plans allows for bias adjustment
while preserving the trend of the model.

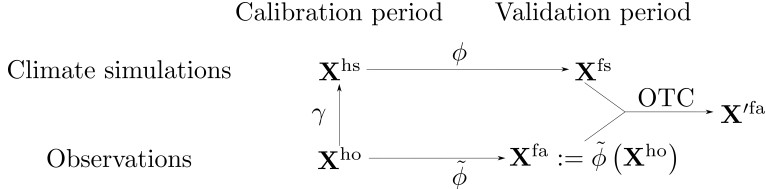

**Figure 2.** The different transport plans and transformations used in dOTC. Based on Robin et al. (2019).

Optimal transport is applied on the basis of an optimal transport plan. The optimal plan between $\mathbf{X}^{\mathrm{ho}}$ and $\mathbf{X}^{\mathrm{hs}}$ is denoted
as $\gamma$. The second optimal plan between $\mathbf{X}^{\mathrm{hs}}$ and $\mathbf{X}^{\mathrm{fs}}$ is denoted as $\phi$. The goal is to transform $\phi$ according to $\gamma$, defining a new
plan $\tilde{\phi}$. This optimal plan estimates $\mathbf{X}^{\mathrm{fa}} = \tilde{\phi}\left(\mathbf{X}^{\mathrm{ho}}\right)$. Finally, $\mathbf{X}^{\mathrm{fs}}$ is adjusted with respect to $\mathbf{X}^{\mathrm{fa}}$, creating the final adjusted
$\mathbf{X}'^{\mathrm{fa}}$. These steps are summarised in Fig. 2.

For the definition of the optimal plan, the 'Optimal Transport Correction' (OTC) (Robin et al., 2019) is used. First, the
empirical distributions $\hat{\mathbf{P}}_{\mathbf{X}^{\mathrm{ho}}}$ and $\hat{\mathbf{P}}_{\mathbf{X}^{\mathrm{hs}}}$ have to be calculated. To achieve this, the subspace of $\mathbb{R}^N$ that contains the data is
partitioned into regularly spaced cells, generically denoted $\mathbf{c}_i^*$, with $N$ the number of variables of $\mathbf{X}^{\mathrm{ho}}$ and $\mathbf{X}^{\mathrm{hs}}$. Using this
notation, $\hat{\mathbf{P}}_{\mathbf{X}^{\mathrm{ho}}}$ and $\hat{\mathbf{P}}_{\mathbf{X}^{\mathrm{hs}}}$ can be estimated using the relative frequencies $p_{\mathbf{X}^{\mathrm{ho}}}$ and $p_{\mathbf{X}^{\mathrm{hs}}}$ as:

$$p_{\mathbf{X}^{\mathrm{ho}},i} = \frac{1}{m} \sum_{k=1}^{m} \mathbf{1}\left(\mathbf{X}_k^{\mathrm{ho}} \in \mathbf{c}_i^*\right), \tag{30a}$$

$$p_{\mathbf{X}^{\mathrm{hs}},i} = \frac{1}{n} \sum_{l=1}^{n} \mathbf{1}\left(\mathbf{X}_l^{\mathrm{hs}} \in \mathbf{c}_i^*\right), \tag{30b}$$

with $\mathbf{1}$ the indicator function and $m$ and $n$ the total number of time steps of respectively $\mathbf{X}^{\mathrm{ho}}$ and $\mathbf{X}^{\mathrm{hs}}$. Thus, the distributions
are essentially estimated by counting the number of observations of each time series within each cell. The optimal plan $\gamma$
between $\mathbf{X}^{\mathrm{ho}}$ and $\mathbf{X}^{\mathrm{hs}}$ can be estimated as:

$$\hat{\gamma} = \sum_{i,j=1}^{I,J} \gamma_{i,j} . \tag{31}$$



The coefficients $\gamma_{i,j}$ are the probabilities to transform an observation of $\mathbf{X}^{\mathrm{ho}}$ in cell $\mathbf{c}_i^*$ into an observation of $\mathbf{X}^{\mathrm{hs}}$ in cell $\mathbf{c}_j^*$ and $I$ and $J$ are the total number of cells containing observations of respectively $\mathbf{X}^{\mathrm{ho}}$ and $\mathbf{X}^{\mathrm{hs}}$. Note that from here on, $\mathbf{c}_i^*$ and $\mathbf{c}_j^*$ denote only those cells containing observations of respectively $\mathbf{X}^{\mathrm{ho}}$ and $\mathbf{X}^{\mathrm{hs}}$ and that 'observation' is used generically for both observed and simulated time series. The coefficients are unknown, but obey the marginal properties:

$$\sum_{j=1}^{J} \gamma_{i,j} = p_{\mathbf{X}^{\mathrm{ho}},i} \tag{32a}$$

and

$$\sum_{i=1}^{I} \gamma_{i,j} = p_{\mathbf{X}^{\mathrm{hs}},j}. \tag{32b}$$

Central to the optimal transport theorem is the cost function $C$ (Villani, 2008), which can here be approximated by

$$\hat{C}(\hat{\gamma}) = \sum_{i,j=1}^{I,J} \| \mathbf{c}_i - \mathbf{c}_j \|^2 \gamma_{i,j}, \tag{33}$$

with $\| \cdot \|$ the Euclidean norm and $\mathbf{c}_i$ and $\mathbf{c}_j$ the centres of the cells defined above. Finding $\gamma_{i,j}$ comes down to solving the
490 problem defined by the constraints of Eqs. (32) and minimisation of Eq. (33). Here, Sinkhorn's algorithm (Cuturi, 2013) is used to find the solution to this problem and thus the optimal plan $\gamma$. Using this optimal plan, a vector of probabilities with length $J$ can be defined for each cell $\mathbf{c}_i^*$ of $\mathbf{X}^{\mathrm{ho}}$, each element $j$ corresponding to the probability that an observation in that cell $\mathbf{c}_i^*$ will be transformed into an observation in cell $\mathbf{c}_j^*$ of $\mathbf{X}^{\mathrm{hs}}$. In the transformation, the vectors of probabilities are used to introduce stochasticity, by sampling from these vectors the element $j$ corresponding to cell $\mathbf{c}_j^*$. The stochastic transformation
of an observation of $\mathbf{X}^{\mathrm{ho}}$ into an observation of $\mathbf{X}^{\mathrm{hs}}$ can be repeated to create an ensemble of results. This ensemble accounts for random weather effects and can thus be considered to be more similar to the true range of observations.

The optimal plan $\phi$ can be calculated analogously. This optimal plan $\phi$ transforms an observation of $\mathbf{X}^{\mathrm{hs}}$ in cell $\mathbf{c}_j^*$ into an observation of $\mathbf{X}^{\mathrm{fs}}$ in cell $\mathbf{c}_k^*$, with $\mathbf{c}_k^*$ defined analogously to $\mathbf{c}_i^*$ and $\mathbf{c}_j^*$. What distinguishes dOTC from OTC is the next phase, in which $\phi$ is transformed according to $\gamma$, resulting in $\tilde{\phi}$. This is conducted in three steps, the first being is the transformation of
500 $\phi$ into a vector. The vector $\mathbf{v}_{jk} := \mathbf{c}_k - \mathbf{c}_j$ represents the climatic trend from an observation of $\mathbf{X}^{\mathrm{hs}}$ in cell $\mathbf{c}_j^*$ to an observation of $\mathbf{X}^{\mathrm{fs}}$ in cell $\mathbf{c}_k^*$. The second step is the transfer according to $\gamma$. The result $\tilde{\phi}$ can be defined by translating the observations of $\mathbf{X}^{\mathrm{ho}}$ along their respective vectors $\mathbf{v}_{jk}$: an observation of $\mathbf{X}^{\mathrm{fa}}$ is then given by $\mathbf{X}_t^{\mathrm{ho}} + \mathbf{v}_{jk}$, with $\mathbf{X}_t^{\mathrm{ho}}$ the observation of $\mathbf{X}^{\mathrm{ho}}$ at time step $t$. However, the translation of $\mathbf{X}_t^{\mathrm{ho}}$ along vector $\mathbf{v}_{jk}$ does not always define an optimal transport plan: the vector has to be adapted to $\mathbf{X}^{\mathrm{ho}}$, which is the third step. In this step, a matrix factor $\mathbf{D}$ is introduced, which rescales the vector $\mathbf{v}_{jk}$. This
rescaling is actually the replacement of the scale of $\mathbf{X}^{\mathrm{hs}}$ by that of $\mathbf{X}^{\mathrm{ho}}$. A Cholesky decomposition of the covariance matrix has been proposed for this rescaling (Bárdossy and Pegram, 2012; Cannon, 2016). Denoting the covariance matrix as $\Sigma$, and the Cholesky decomposition as $\mathrm{Cho}(\Sigma)$, Robin et al. (2019) proposed to multiply $\mathbf{v}_{jk}$ by the following matrix:

$$\mathbf{D} := \mathrm{Cho}(\Sigma_{\mathbf{X}^{\mathrm{ho}}}) \cdot \mathrm{Cho}(\Sigma_{\mathbf{X}^{\mathrm{hs}}})^{-1}. \tag{34}$$





Robin et al. (2019) remark that the Cholesky decomposition only exists if $\Sigma$ is symmetric and positive-definite. Some covari-
510 ance matrices, such as those of highly correlated random variables, do not have this property. $\Sigma$ must then be slightly perturbed
to be positive-definite (Higham, 1988; Knol and ten Berge, 1989). It is also possible for the Cholesky decomposition to be
poorly estimated if the available data are too small compared to the dimension. In that case, it is suggested to replace the matrix
$\mathbf{D}$ by the diagonal matrix of the standard devation: $\mathbf{D} = \mathrm{diag}\left(\sigma_{\mathbf{X}^{\mathrm{ho}}}\sigma_{\mathbf{X}^{\mathrm{hs}}}^{-1}\right)$.

An observation of $\mathbf{X}^{\mathrm{fa}}$ is then given by $\mathbf{X}_t^{\mathrm{ho}} + \mathbf{D} \cdot \mathbf{v}_{jk}$. To finalize, the empirical distribution $\hat{\mathbf{P}}_{\mathbf{X}^{\mathrm{fa}}}$ can be calculated. Using
this distribution, OTC can be applied to $\mathbf{X}^{\mathrm{hs}}$ and $\mathbf{X}^{\mathrm{fa}}$ to generate $\mathbf{X}^{\prime\mathrm{fa}}$. A more elaborate mathematical explanation can be
found in Robin et al. (2019), a summary is given in Algorithm 4.

---

**Algorithm 4** dOTC

**Input:**

    Historical observations $\mathbf{X}^{\mathrm{ho}}$

    Historical simulations $\mathbf{X}^{\mathrm{hs}}$

    Future simulations $\mathbf{X}^{\mathrm{fs}}$

**Output:**

    Adjusted future simulations $\mathbf{X}^{\prime\mathrm{fa}}$

    Calculate the empirical distributions $\hat{\mathbf{P}}_{\mathbf{X}^{\mathrm{ho}}}$, $\hat{\mathbf{P}}_{\mathbf{X}^{\mathrm{hs}}}$ and $\hat{\mathbf{P}}_{\mathbf{X}^{\mathrm{fs}}}$

    Calculate the optimal plan $\gamma$ between $\hat{\mathbf{P}}_{\mathbf{X}^{\mathrm{ho}}}$ and $\hat{\mathbf{P}}_{\mathbf{X}^{\mathrm{hs}}}$

    Calculate the optimal plan $\phi$ between $\hat{\mathbf{P}}_{\mathbf{X}^{\mathrm{fs}}}$ and $\hat{\mathbf{P}}_{\mathbf{X}^{\mathrm{hs}}}$

    Calculate the Cholesky factor $\mathbf{D}$

    **for** all $\mathbf{X}_t^{\mathrm{ho}}$ **do**

        Find cell $\mathbf{c}_i$ containing $\mathbf{X}_t^{\mathrm{ho}}$

        Construct the vector of probabilities $\hat{\gamma}_{\mathbf{X}_t^{\mathrm{ho}}} = (\gamma_{i,1}, \ldots, \gamma_{i,J})/p_{\mathbf{X}^{\mathrm{ho}},i}$

        Sample $j \in \{1, ..., J\}$ according to the probability vector $\hat{\gamma}_{\mathbf{X}_t^{\mathrm{ho}}}$

        Construct the vector of probabilities $\hat{\phi}_{\mathbf{X}_t^{\mathrm{hs}}} = (\phi_{j,1}, \ldots, \gamma_{j,K})/p_{\mathbf{X}^{\mathrm{hs}},j}$

        Sample $k \in \{1, ..., K\}$ according to the probability vector $\hat{\phi}_{\mathbf{X}_t^{\mathrm{hs}}}$

        Calculate the vector $\mathbf{v}_{jk}$

        Calculate $\mathbf{X}_t^{\mathrm{fa}} = \mathbf{X}_t^{\mathrm{ho}} + \mathbf{D} \cdot \mathbf{v}_{jk}$

    **end for**

    Calculate the empirical distribution $\hat{\mathbf{P}}_{\mathbf{X}^{\mathrm{fa}}}$

    Apply OTC between $\mathbf{X}^{\mathrm{fa}}$ and $\mathbf{X}^{\mathrm{fs}}$ to generate $\mathbf{X}^{\prime\mathrm{fa}}$

---



### 3.4 Experimental design

Prior to all intensity-bias-adjusting methods, the thresholding occurrence-adjusting method was applied. In the intensity-bias-adjustment step, a balance was sought between randomness and computational power for the calculation of the intensity-bias-adjusting methods. Methods with randomised steps were repeated. As such, 10 calculations were made for dOTC. The resulting values of each index were averaged for further comparison. Biases on the indices were always calculated as raw or adjusted simulations minus observations, indicating a positive bias if the raw or adjusted simulations are larger than the observations and a negative bias if the simulations are smaller.

## 4 Results

In this section, the results will be shown first for the R index calculations for bias change, and then for the validation indices. For the validation indices, first the indices based on the adjusted variables are discussed, followed by an elaboration on the indices based on the derived variables. As the effect on discharge is the overarching goal of this paper and the discharge indices are affected by all other indices, those will be discussed last. All observed values and biases of both raw and adjusted simulations are presented in Table A1.

### 4.1 Bias change

The R index values for the variable averages, standard deviations and all indices are given in Table 2. The results vary considerably depending on the variable and/or index: for P, the bias can be considered almost stationary: only the 99.5th percentile has an R index above one. In contrast, for E, the R index values are above 1 for the middle percentiles and for the standard deviation, indicating some major changes in parts of the distribution, and consequently, the bias. For T, the mean and the lower extremes are clearly influenced, although the bias on the higher extremes does not change. The different effects on the variables are linked with the effect on the (cross-)correlations. For example, the lag-1 cross-correlation between P and E has an R index value of only 0.19, whereas the R index value for the cross-correlation between E and T is 1.20. Although the R index values are low for P, this does not imply that the R index values for the precipitation occurrence indices are low. With an R index value of 1.44, the auto-correlation bias clearly changes between both periods. However, this is not reflected by the other precipitation occurrence indices, which all but one have R index values lower than one.

Many of the R index values presented in Table 2 indicate that the bias changes between the two periods considered here (1970-1989 versus 1998-2017) might already be large enough to have an effect on the bias adjustment. As these periods are only separated by 10 years, this is an important indicator for the bias adjustment of late 21st century data, just as Chen et al. (2015) mentioned. However, it does not suffice to calculate just a few of these R index values. The results vary substantially among variables and even for the percentiles of a variable under consideration: while the 5th T percentile has an R index value of 2, the value for the 95th percentile is only 0.07. This could give an indication of why the methods perform more poorly for some of these indices. However, purely based on these results, it is impossible to say exactly what causes the





**Table 2.** R index values for 1970-1989 as historical period and 1998-2017 as future period

| Precipitation | | Temperature | | Pot. Evaporation | | Occurrence & Correlation | |
|---|---|---|---|---|---|---|---|
| **Indices** | **R index** | **Indices** | **R index** | **Indices** | **R index** | **Indices** | **R index** |
| $P_5$ | NaN | $T_5$ | 2 | $E_5$ | 0.03 | $P_{lag1}$ | 1.44 |
| $P_{25}$ | 0 | $T_{25}$ | 2 | $E_{25}$ | 0.47 | $P_{P00}$ | 0.09 |
| $P_{50}$ | 0.10 | $T_{50}$ | 2 | $E_{50}$ | 1.47 | $P_{P10}$ | 0.41 |
| $P_{75}$ | 0.13 | $T_{75}$ | 0.87 | $E_{75}$ | 2 | $N_{dry}$ | 0.29 |
| $P_{90}$ | 0.19 | $T_{90}$ | 0.31 | $E_{90}$ | 1.14 | $corr_{E,T}$ | 0.75 |
| $P_{95}$ | 0.17 | $T_{95}$ | 0.07 | $E_{95}$ | 1 | $corr_{P,E}$ | 0.20 |
| $P_{99}$ | 0.58 | $T_{99}$ | 0.19 | $E_{99}$ | 0.47 | $corr_{P,T}$ | 0 |
| $P_{99.5}$ | 1.02 | $T_{99.5}$ | 0.08 | $E_{99.5}$ | 0.20 | $crosscorr_{E,T,0}$ | 2 |
| Pmean | 0.18 | Tmean | 2 | Emean | 1.06 | $crosscorr_{E,T,1}$ | 0.90 |
| Pstd | 0.72 | Tstd | 0.50 | Estd | 2 | $crosscorr_{P,E,0}$ | 0.31 |
| | | | | | | $crosscorr_{P,E,1}$ | 0.13 |
| | | | | | | $crosscorr_{P,T,0}$ | 0.10 |
| | | | | | | $crosscorr_{P,T,1}$ | 0.09 |

bias nonstationarities. Possible causes could be that recent trends such as those in precipitation extremes (Papalexiou and Montanari, 2019) are poorly captured by the models, that limiting mechanisms such as soil moisture (Bellprat et al., 2013) are

poorly modelled or that natural variability influences (Addor and Fischer, 2015) the biases. However, discussing this in depth is out of the scope of the present study and deserves a separate study. In what follows, we will focus on the performance of the bias-adjusting methods and whether or not there is a link with these nonstationarities.

## 4.2 Precipitation amount

Figure 3 presents the $RB_O$ and $RB_{MB}$ values for the highest P percentiles. None of the residual bias values of the lower

percentiles can be plotted as either the observations are 0 mm ($P_5$ and $P_{25}$) or the $RB_O$ values are lower than zero ($P_{50}$). The percentiles could also have been plotted for wet days only (e.g. days with P higher than 0.1 mm/day), but as some methods change the number of dry days after the initial thresholding step, the dry days are also included in the calculation of the indices. This influences the $RB_O$ and $RB_{MB}$ values: they are generally higher when the dry days are not included.

The $RB_O$ and $RB_{MB}$ values depict a very similar performance for QDM, mQDM and MBCn, but a different performance

for MRQNBC and dOTC. The similar performance of the former three is unsurprising, as their adjustments of P are all very similar. The only difference between QDM and MBCn versus mQDM is the time series to which the adjustment was applied, as the latter is based on the historical time series. QDM, mQDM and MBCn are consistently the best methods out of the five tested here, with the $RB_{MB}$ values for $P_{75}$, $P_{90}$, $P_{95}$ and $P_{99}$ all below 0.5 and the $RB_O$ values also below 0.5. The performances





of MRQNBC and dOTC are worse, but not poor either: $P_{75}$, $P_{90}$, $P_{95}$ and $P_{99}$ all have $RB_O$ and $RB_{MB}$ values lower than 1, but
only for dOTC the majority of them ($P_{75}$, $P_{90}$ and $P_{95}$) are below 0.5 for $RB_{MB}$. For both MRQNBC and dOTC, no $RB_O$ values
below 0.5 are obtained. As seen in Section 4.1, P was one of the few indices having almost all R index values below 1. This
might be linked to the generally good results for P illustrated in this section.

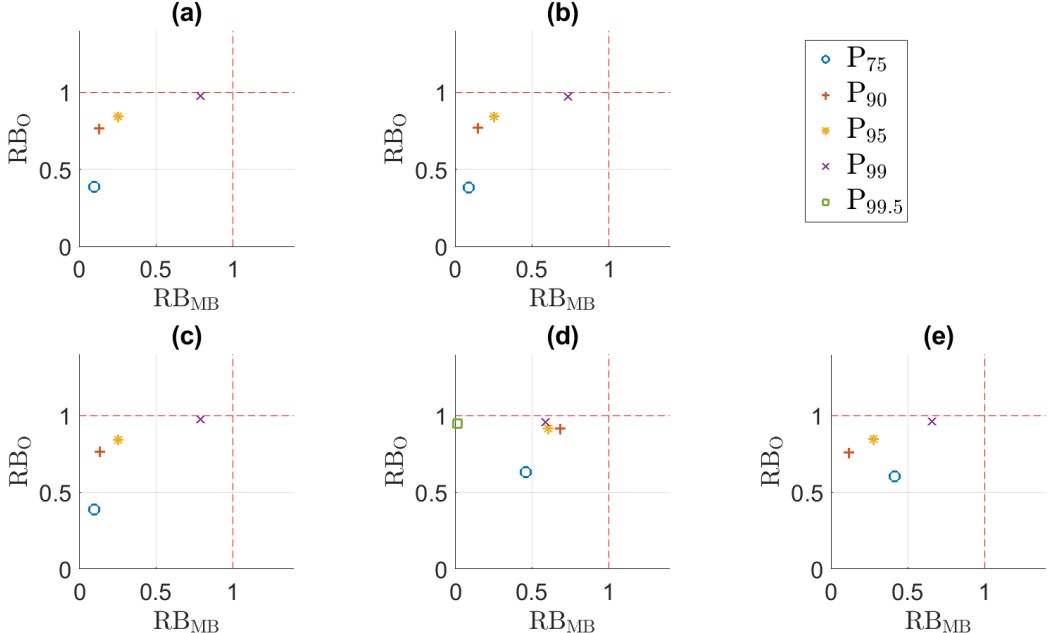

**Figure 3.** $RB_{MB}$ versus $RB_0$ for the precipitation indices. (a) QDM, (b) mQDM, (c) MBCn, (d) MRQNBC, (e) dOTC.

A surprising result for P is the high $RB_{MB}$ value for $P_{99.5}$ for MRQNBC. This percentile is too biased for all other methods
to be plotted. As the R index value was close to 1, it is possible that bias nonstationarity slightly influences the performance.
For MRQNBC, however, the combination of QDM with the focus on correlation seemingly improves the performance of this
percentile. As heavy precipitation values are clustered in time, the performance of the respective indices might be improved by
the correlation. The good representation of heavy precipitation values in the MRQNBC-adjusted time series is also shown by
$P_{99}$, for which MRQNBC has an $RB_{MB}$ value of 0.59, the best of all methods.

### 4.3    Temperature

For the temperature adjustment, the $RB_O$ and $RB_{MB}$ values indicate that all methods result in a performance better than the
raw climate simulations, except for MRQNBC (Fig. 4). In contrast to all other methods, only the residual bias values of $T_{90}$ of
MRQNBC are within the area delineated by the 1-1-lines. For all other indices, the bias is worse than in the climate simulations,
with absolute biases up to 7ºC. The results for MRQNBC are interesting, as T is the best understood variable (Shepherd, 2014)





and should thus not be hard to adjust. One line of reasoning could be the implementation of model trend preservation. While

trend preservation is a prominent aspect in all other methods, the persistence preservation is at least as important in MRQNBC. The trade-off between both aspects of the bias adjustment thus seems to influence the result of MRQNBC, while the other methods can more easily adapt to and adjust the simulated T.

When comparing the indices for the other methods, the results are rather similar. They all have $RB_O$ values close to 1, indicating that the bias difference is small in comparison to the absolute T values. Besides that, for every method, the lower T

percentiles have $RB_{MB}$ values that are too high to be plotted. However, despite their similar behaviour, the methods show some notable differences. The highest percentiles have the lowest $RB_{MB}$ values for QDM and MBCn, which have the same percentiles by construction, but this differs for the other methods. For example, $T_{99.5}$ has an $RB_{MB}$ value of 0.09 for QDM and MBCn, but only 0.43 for dOTC and 0.77 for mQDM. On the other hand, when considering all plotted percentiles, dOTC generally performs best. The highest $RB_{MB}$ value for dOTC is 0.52 ($T_{75}$), whereas 0.65 ($T_{75}$) is the highest value for QDM

and MBCn and 0.77 ($T_{99.5}$) for mQDM. Although broadly similar, the indices for QDM and mQDM display some interesting differences. Whereas for QDM $T_{95}$, $T_{99}$ and $T_{99.5}$ have the lowest $RB_{MB}$ values, $T_{75}$ has the lowest value for mQDM. In contrast, QDM has the highest $RB_{MB}$ value for $T_{75}$. This might imply that for the highest T values, it is better to follow the simulations, while for slightly lower values, it is better to only use the climate change signal. Yet, QDM has the best $RB_{MB}$ values and might thus be preferable.

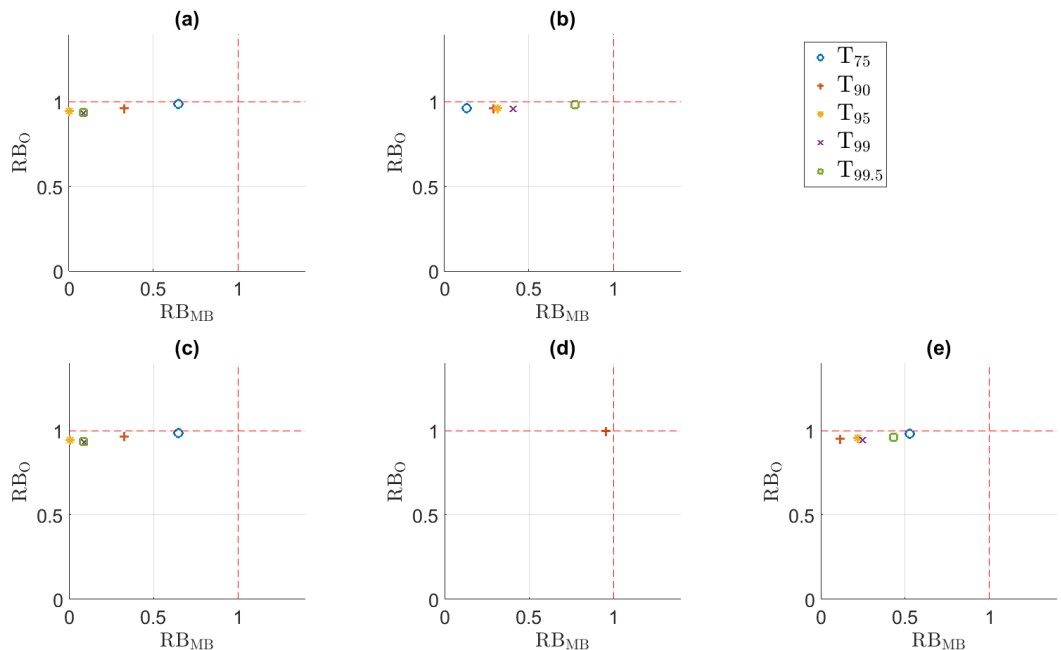

**Figure 4.** $RB_{MB}$ versus $RB_0$ for the temperature indices. (a) QDM, (b) mQDM, (c) MBCn, (d) MRQNBC, (e) dOTC.





For the lowest T values, all methods seem unable to handle the change in bias (as seen in Table 2): the $RB_{MB}$ values are all higher than 1. This poor performance, combined with the high values for $RB_O$, might imply that it is better not to adjust T and work with the raw climate simulations. However, for the extreme T values, the absolute biases can be more than 1°C. Thus, depending on the research goal and the R index value, it might be important to consider whether or not T should be adjusted.

### 4.4  Potential evaporation

Figure 5 displays the $RB_O$ and $RB_{MB}$ values for the E indices. Only a few indices are shown for each method, or just one for dOTC, indicating that the performance after adjusting the bias is generally worse than the raw climate simulations. The indices plotted are $E_{25}$, $E_{99}$ and $E_{99.5}$. The index $E_5$ also performs well, but cannot be plotted as its observed value is 0 mm. Thus, for the lowest and the highest percentiles, the bias-adjusting methods perform well, but they fail to capture the nonstationarity at the middle percentiles. These middle percentiles have high R index values: they are all greater than or equal to one. Only for

dOTC, it is possible to plot a percentile between $E_{25}$ and $E_{99}$: $E_{95}$. However, for dOTC, this is also the only percentile for which it is possible to plot the $RB_O$ and $RB_{MB}$ values (respectively 1.00 and 0.86). For all other methods and all other percentiles, both $RB_O$ and $RB_{MB}$ values are higher than 1. The poor performance of dOTC might be related to its trend preservation characteristics. Of all methods, it is the one most explicitly designed to follow the simulated trend. This might thus imply that the nonstationarity for E is caused by poor model performance, although this should be investigated more in depth.

The percentiles that are plotted all have a high $RB_O$ value, which is in this case caused by rather low biases to adjust. For example, $E_{99.5}$ had an observed value of 5.24 mm/day and only a bias of 0.27 mm/day, or 5%. There is consequently not much room for improvement, though the $RB_{MB}$ values imply that the bias-adjusting method could be improved, except for $E_{99.5}$, which consistently has an $RB_{MB}$ value lower than 0.5.

As in Section 4.3, there are interesting differences between QDM and mQDM. In contrast to the results for temperature,
mQDM performs better. For mQDM, the highest percentiles ($E_{99}$ and $E_{99.5}$) have lower $RB_{MB}$ values than for QDM. In this case, it thus seems better to only use the climate change signal. However, given the general poor performance of all methods, these results should be considered with care.

Given that the percentiles with a high R index value have a larger bias than the raw simulations after adjustment, and the added value for the other percentiles with respect to observed values is low, it can be advised not to adjust E. However, similar
to T, this should be evaluated on a case-by-case basis.

### 4.5  Correlation

When considering the correlation (Fig. 6), the methods generally perform well: most of the correlation indices can be plotted. However, there are some differences depending on the indices under consideration and the method. The indices that can always be plotted are the lag-0 cross-correlation between P and T, the lag-1 cross-correlation between P and T, the lag-1
cross-correlation between P and E and the correlation between P and T. Except for dOTC, all methods also perform well for the lag-0 cross-correlation between P and E. Yet, for the indices that can be plotted, the $RB_O$ and $RB_{MB}$ values show a considerable difference among the methods. For example, the lag-1 cross-correlation between P and E has $RB_O$ values ranging





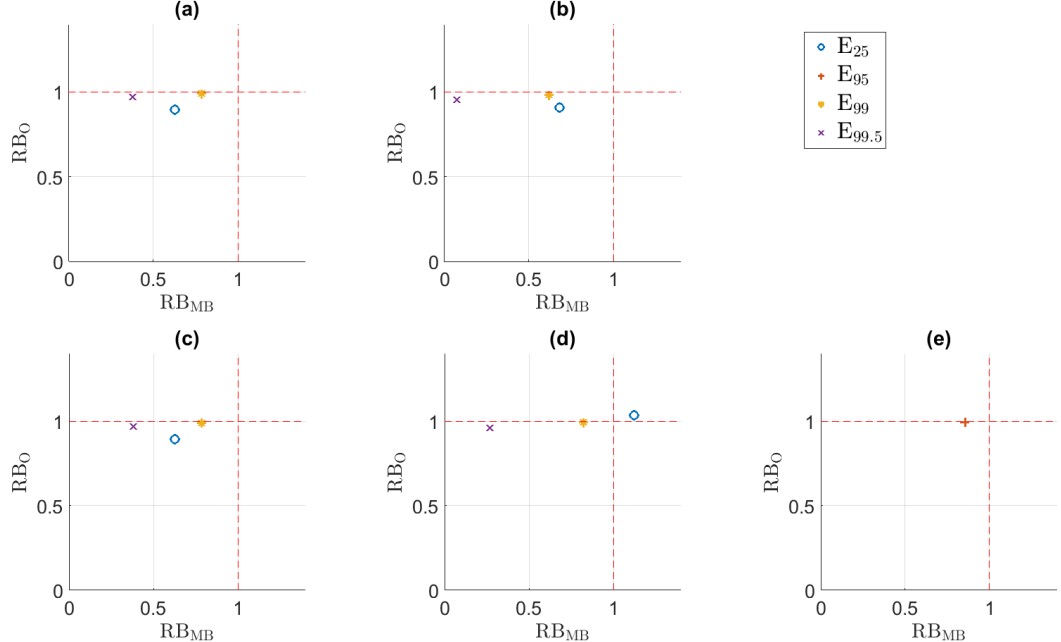

**Figure 5.** $RB_{MB}$ versus $RB_0$ for the potential evaporation indices. (a) QDM, (b) mQDM, (c) MBCn, (d) MRQNBC, (e) dOTC.

from 0.29 (mQDM) to 0.69 (dOTC) and $RB_{MB}$ values ranging from 0.08 (mQDM) to 0.60 (dOTC). The best method varies for each index: while dOTC does not perform well for most indices, it has the best performance for the correlation between P

and T. It is interesting that the performance of dOTC seems to be either very good, or very poor. As dOTC is built around the idea of trend preservation and all-in-one adjustment, it is possible that the adjustment performs very well when the trend is properly modelled. Three out of the four correlations that dOTC adjusts well are based on T and P, the two variables that are well understood in the time frame under consideration. All indices that are not or less frequently present in the plots have one thing in common: they are based on the correlation between E and one of the other variables. The indices based on E thus

consequently perform worst, except for the aforementioned lag-1 cross-correlation between P and E.

    The correlation index performance seems to be related to the results of T (Section 4.3) and E (Section 4.4): correlations of E with another variable generally perform worse, and the (cross-)correlation between T and E with another variable performs the worst, in line with the R index values (Table 2). Although the multivariate bias-adjusting methods are supposed to adjust correlation, they seem to be unable to do so, as they generally have larger biases (though not on all indices) than the univariate

bias-adjusting methods for the indices with the lowest residual bias values. This seems to indicate that the multivariate bias-adjusting methods, and especially MBCn and dOTC, are unable to adjust the correlation exactly because of the nonstationarity in the correlation that has to be overcome. In contrast, the univariate bias-adjusting methods neglect the adjustment of correlation and consequently do not have to overcome nonstationarity in the correlation bias. Yet, for the univariate bias-adjusting





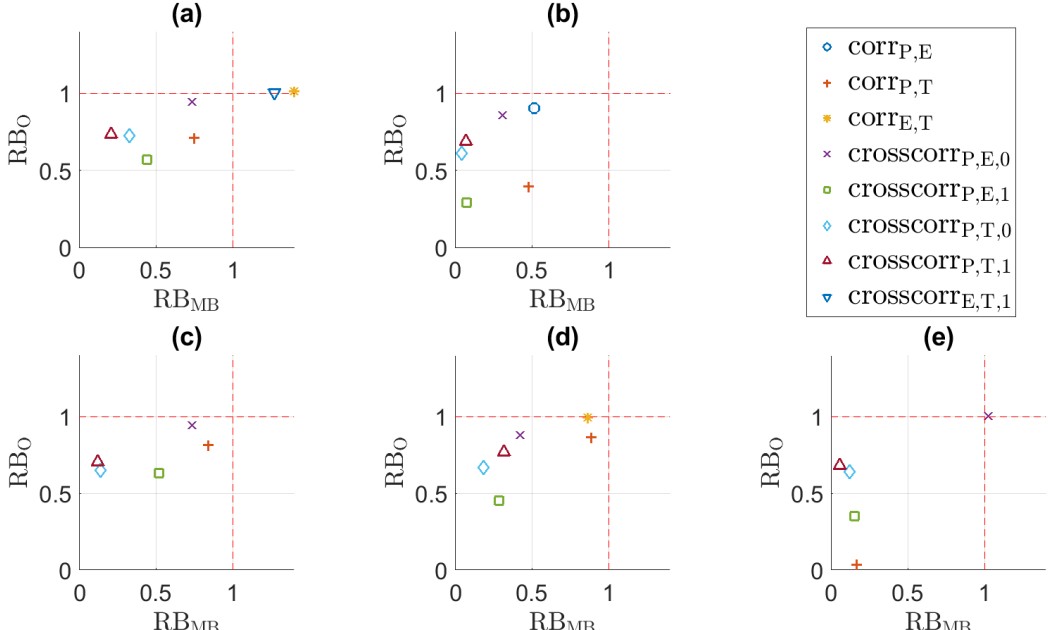

**Figure 6.** $RB_{MB}$ versus $RB_O$ for the correlation indices. (a) QDM, (b) mQDM, (c) MBCn, (d) MRQNBC, (e) dOTC.

methods, the difference in adjustment of T and E seems to have an influence here as well, as illustrated by the different results

for QDM and mQDM. Except for the correlations that have high $RB_{MB}$ values for all methods, the results indicate that mQDM performs better. Thus, it might be better to use correlations of the observed time series than to adjust the simulated correlations. This is confirmed by the results for MRQNBC. Together with mQDM, this is the only method to have six indices with $RB_{MB}$ values lower than one. Besides, it is also the only multivariate bias-adjusting method to have an $RB_{MB}$ value for $corr_{E,T}$ lower than one, although it is only slightly lower.

**4.6 Precipitation occurrence**

Figure 7 displays the $RB_O$ and $RB_{MB}$ values for the precipitation occurrence indices. When comparing these values, large differences among the methods and among the indices can be noted. The best-performing method seems to be QDM, as all the $RB_{MB}$ values are lower than 0.5 and some $RB_O$ values are close to 0.5. When comparing the other methods, there is no clear difference between the univariate and multivariate bias-adjusting methods. The other univariate method, mQDM, and one

multivariate method, MRQNBC, also perform better than the raw climate simulations for all indices. The other two methods, MBCn and dOTC, have respectively only one and two indices with both $RB_O$ and $RB_{MB}$ values below 1.

Interestingly enough, the indices with $RB_O$ and $RB_{MB}$ values below 1 are not the same for all methods. For the three best-performing methods, the dry-to-dry transition probability has very low $RB_{MB}$ values (ranging from 0.3 to 0.18), while this



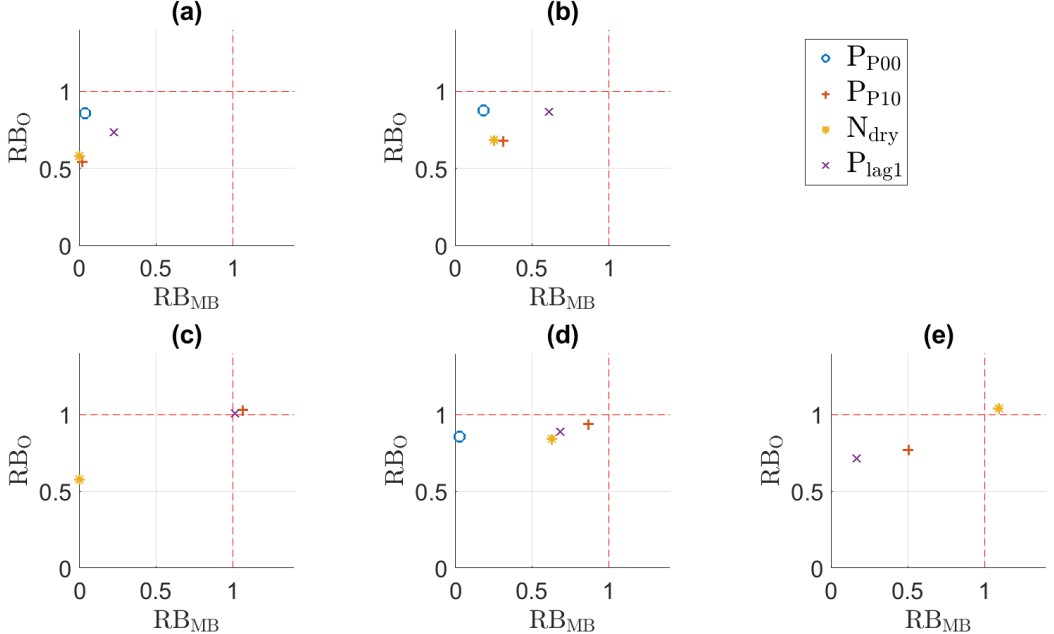

**Figure 7.** $RB_{MB}$ versus $RB_0$ for the precipitation occurrence indices. (a) QDM, (b) mQDM, (c) MBCn, (d) MRQNBC, (e) dOTC.

index is absent from the MBCn and dOTC plots. The differences between those two plots are also notable. For MBCn, only the

number of dry days has a very low $RB_{MB}$ value (0), as the number of dry days is unaffected after the thresholding, whereas the

lag-1 P auto-correlation and the wet-to-dry transition probability are more biased than the raw climate simulations. For dOTC,

it is the other way around: the number of dry days is more biased after the application of dOTC, and the auto-correlation and

the wet-to-dry transition probability perform well, with $RB_{MB}$ values 0.50 or lower.

     Another peculiar result that can be seen from Fig. 7 is the difference in dry day bias. Although all methods start with the

same number of dry days, there are large differences among the $RB_O$ and $RB_{MB}$ values for the number of dry days. The $RB_O$

values range from 0.58 to 1.04 and the $RB_{MB}$ values from 0 to 1.09. QDM and MBCn perform best ($RB_{MB} = 0$), as the number

of dry days is unaffected after the thresholding. For mQDM ($RB_{MB} = 0.25$), this holds by construction: instead of adjusting the

threshold-adjusted climate model simulations, this method adjusts the observations. For MRQNBC ($RB_{MB} = 0.63$) and dOTC

($RB_{MB} = 1.09$), the results seem to imply that the multivariate framework of these methods has an influence on the number of

dry days.

     What the difference in transition probabilities implies for the time series, becomes more clear in Fig. 8. Although all adjusted

simulations and the observations have more short wet spells than long ones, MBCn pronounces the short wet spell length more

than the other methods, while the probability of longer wet spell lengths is lowered in comparison with other methods. Closest

to the observations is mQDM. QDM and MRQNBC also perform well, a conclusion similar to that of Fig. 7.





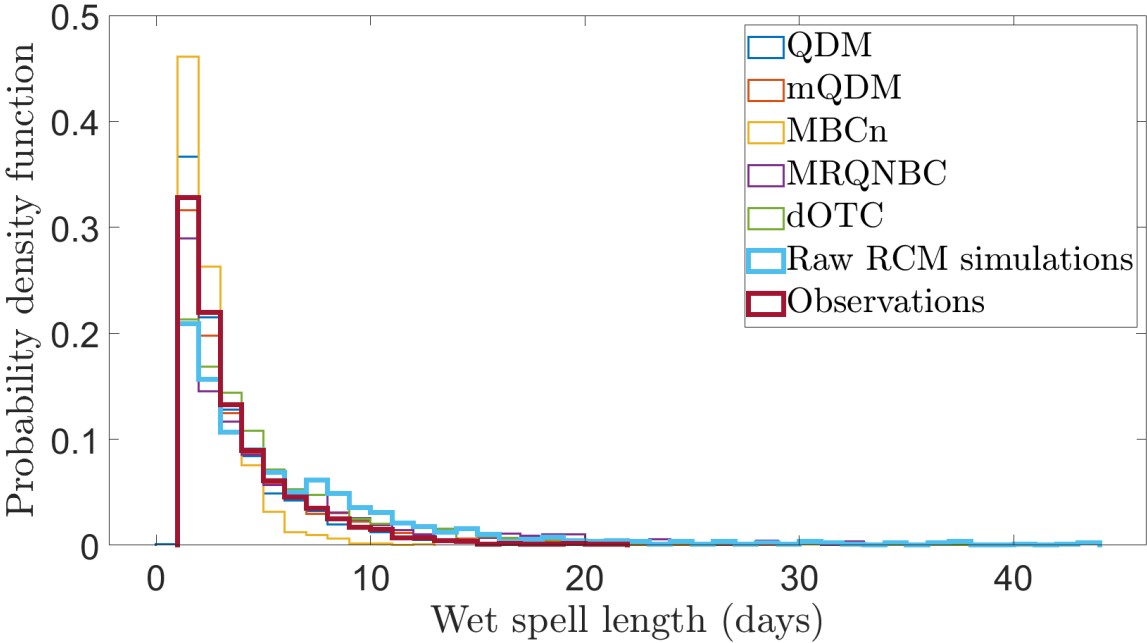

**Figure 8.** Wet spell length probability mass function for all adjusted simulations, the raw RCM simulations and the observations.

The difference in performance between QDM and the other methods seems to demonstrate that most strategies for retaining a certain temporal structure or adjusting the temporal structure do not perform well. MRQNBC and mQDM depend heavily on the temporal structure of the observations, and MBCn and dOTC have an important shuffling or recalculation aspect, all of which lead to less reliable results at the end of the process. The poor performance of dOTC and MBCn for the temporal structure was also discussed by François et al. (2020). As for mQDM and MRQNBC, it is notable that the temporal structure

does not change much from the calibration time series to the validation time series. At least, this is suggested by their relatively good performance, which is based on using the observed time series (mQDM) and observed persistence statistics (MRQNBC). Yet, this is no guarantee that these methods will be able to realistically adjust climate model simulations for the end of the century.

Figure 7 also suggests that despite the high R index value, the P lag-1 auto-correlation is not necessarily poorly adjusted.

For QDM, this index has relatively low $RB_O$ and $RB_{MB}$ values. This could imply that the performance still depends on the robustness to the bias nonstationarity of the methods under consideration. Or, as the other indices illustrate, the effect of bias (non-)stationarity is not as large as the effect induced by the methods themselves. An example of this is the number of dry days: though it has a low R index value, the performance varies substantially among the methods.





## 4.7 Discharge

All bias-adjusting methods perform better for the discharge percentiles compared to most other indices (Fig. 9). Although the discharge is influenced by a combination of many effects, these appear to be small in the end result. For example, the poor performance of E (Fig. 5) does not result in large discharge biases. Thus, it is the integration of precipitation amount, precipitation occurrence and evaporation, and the routing effect that ultimately defines the resulting discharge. Generally, many indices perform well, though there are some differences among the methods. The best-performing methods are QDM

and mQDM, as all indices have $RB_{MB}$ values lower than 0.5 and some indices have $RB_O$ values near to zero. For mQDM, the 20-year return period even has an $RB_O$ value of -0.05. For MBCn, the results are also good. Most extreme values have $RB_O$ and $RB_{MB}$ values lower than 0.5; only the 5th percentile has an $RB_O$ value of 0.96 and an $RB_{MB}$ value of 0.89. As the only difference between QDM and MBCn is the adjustment of occurrence, the results for discharge illustrates the importance of occurrence adjustment. This variability in values seems to be a difference between the univariate and multivariate bias-adjusting methods.

For the two worst performing methods, i.e. MRQNBC and dOTC, some indices have $RB_O$ and $RB_{MB}$ values close to or higher than 1 and some values between 0.5 and 1. These methods are thus unable to correctly adjust the bias for all indices. However, although MRQNCB and dOTC seem to perform similarly, the indices with the worst $RB_O$ and $RB_{MB}$ values are different. For MRQNBC, the 99.5th percentile and the 20-year return period have the highest values, whereas for dOTC, the 5th and 25th percentile perform worse than the raw climate simulations. From the point of view of extreme discharges, dOTC is thus the

better method of these two. This might indicate that although not all occurrence indices of dOTC had lower $RB_{MB}$ values than those of MRQNBC, those that had ($P_{lag1}$ and $P_{P10}$), had a larger influence on the extreme discharge values. Both indices are partly based on the occurrence of wet days, and thus indicate that those need to be at the correct place in the time series for extreme floods to be correctly simulated.

## 5 Discussion

In the previous section, the results for the bias adjustment by different methods and under climate change conditions were reported. The effect of climate change on the bias was evaluated through the R index, which showed that the bias for some indices cannot be considered stationary. For some of the indices (the lower percentiles of T and especially the middle percentiles of E) the methods performed poorly, which could often be linked with the R index values. The methods clearly handle this bias nonstationarity differently. It seems that the univariate bias-adjusting methods are far more robust: even for indices with

high R values, they are sometimes able to perform very well, with low $RB_O$ and $RB_{MB}$ values. This good performance thus seems to imply that the more indices a bias-adjusting method directly adjusts, the more susceptible it is to problems related to bias nonstationarity. However, this does not imply that QDM and mQDM are similar: while they are almost as good for many variables, the poorer performance of mQDM for the precipitation occurrence indices is an indication that assuming that the temporal structure of the past can be used for the future might be dangerous, as Johnson and Sharma (2011) and Kerkhoff et al.

(2014) already mentioned. Given that mQDM performed worse for two time periods separated by 10 years only, it is unlikely that it is safe to use this method, or other delta change-based methods, for impact assessments targeting the end of the 21st





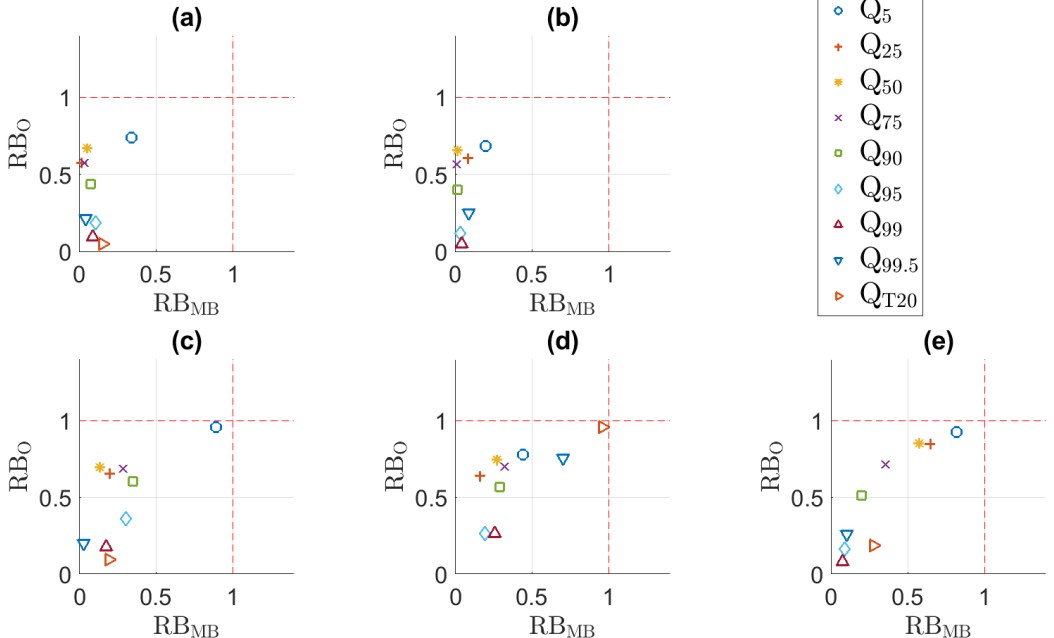

**Figure 9.** $RB_{MB}$ versus $RB_O$ for the discharge percentiles and the 20-year return period value. (a) QDM, (b) mQDM, (c) MBCn, (d) MRQNBC, (e) dOTC.

century and depending on the temporal structure of time series. Yet, for some other indices, especially the correlation, mQDM performed better. Consequently, the exact choice should depend on the goals of the end user.

The results of the multivariate bias-adjusting methods too are not without nuance: though they are generally worse than the
univariate bias-adjusting methods, their performance depends heavily on the variable under consideration and on the method itself. A clear example of this dependence on variables is the contrasting performance of dOTC to adjust T (Fig. 4) versus E (Fig. 9): the adjustment of E is much worse. This is a reminder that in a multivariate context, the multivariate methods are far less robust and can perform relatively good and poor at the same time for different variables. Therefore, there seems to be an interplay between the modelling of the variables and the method of calculation. Except for P, for which the results were
similar, the methods performed differently for each variable. MRQNBC performed best in the context of temporal structure, for which it was designed (Fig. 7). For T, MBCn and dOTC performed better (Fig. 4). This could be related to their trend-preserving properties, which are more pronounced for those methods than for MRQNBC. For E (Fig. 5) and correlation (Fig. 6), dOTC displayed the most different results. For the former, the all-in-one and trend-preservation method did not seem robust enough. For the latter, it depended heavily on the type of correlation under consideration. These results seem to imply that the
difference under bias-nonstationary conditions is not clear-cut for the different types of multivariate bias-adjusting methods. For the 'marginal/dependence' vs. the 'all-in-one' approach, consequently no clear conclusions can be drawn. For the amount of temporal alteration, it depends on the index under consideration. MRQNBC, which replaces the simulated correlations by



those of the observations performs well for the temporal structure, but performs worse for many other indices. For MBCn and dOTC, the effect of the difference in temporal alteration is less distinct and other properties, such as trend preservation, seem

to have more influence.

To have a better view of how these results should be interpreted, the perspective of the end user should be considered (Maraun et al., 2015; Maraun and Widmann, 2018b). We used discharge as an example, using the relatively simple PDM. Although the residual bias values for all methods for E (Fig. 5) indicate a poor performance, the influence thereof on the discharge seems to be negligible (Fig. 9). Discharge is the variable that is of the highest importance for hydrological impact modelling, and the

results indicate that most methods are able to adjust the forcing variables sufficiently in order to have a good simulation of discharge. However, the small differences between the methods should still be taken into account. Overall, QDM and mQDM perform best in adjusting the variables such that the discharge rates are the least biased in comparison with the observations. This is also important considering that bias adjustment can be applied for many different types of impact assessment. In other impact assessments, the differences could affect the result more than the discharge considered here. For example, forest fires

(a typical compound event, discussed in a bias adjustment context in e.g. Yang et al. (2015), Cannon (2018), Zscheischler et al. (2019)) depend more heavily on T and E to simulate fire weather conditions. Besides such compound events, other applications are ecosystem functioning (Sippel et al., 2016), agriculture (Galmarini et al., 2019), or climate zone classification (Beck et al., 2018). In such studies the effect of bias nonstationarity can even be worse, whereas in other studies, depending more on P or other (so far) less-affected variables, the need for a bias nonstationarity-proof bias-adjusting method is less compelling.

Anyway, the inability of some methods to adjust the biases in nonstationary conditions implies that a thorough assessment of possible bias nonstationarity should be made before bias-adjusting. If not done, the risk of reporting a wrong future projection is likely increased. Given the knowledge of bias nonstationarity, such uncertainties can be better characterised.

Returning to the discharge, it might be interesting to discuss whether or not the adjustment of E is truly needed. On the one hand, this variable is the most affected by bias nonstationarity. On the other hand, discharge is far less influenced by this

variable than by P or temporal structure. The discharge has been calculated for this setting with raw E, the result of which is shown in Fig. 10. The results depend on the method: for QDM and mQDM, raw E data slightly exacerbate the results, while for dOTC the percentiles are all improved. Only for MRQNBC and MBCn, the results are highly dependent on the considered percentile. For MBCn, the 5th percentile and the 20-year return period value (with $RB_O \leq 0$) are improved, whereas the 95th and 99th percentile $RB_O$ and $RB_{MB}$ values are deteriorated. For MRQNBC, the results are opposite: the 5th percentile $RB_O$

and $RB_{MB}$ values are deteriorated, and the 95th and 99th values are improved.

The results for raw E seem to imply that, on the one hand and depending on the bias-adjusting method used, a well-considered choice of variables to adjust can give optimal results. On the other hand, the results demonstrate once more that the univariate methods are far more robust than the multivariate methods. Although the $RB_O$ and $RB_{MB}$ values are slightly deteriorated for QDM and mQDM in comparison with the discharge based on adjusted E, all values, and especially the values for the highest

percentiles, still indicate a good bias adjustment. In general, an assessment like this can be done for the other types of impact studies discussed above, so that the influence of adjusting bias-nonstationary variables can be better understood.





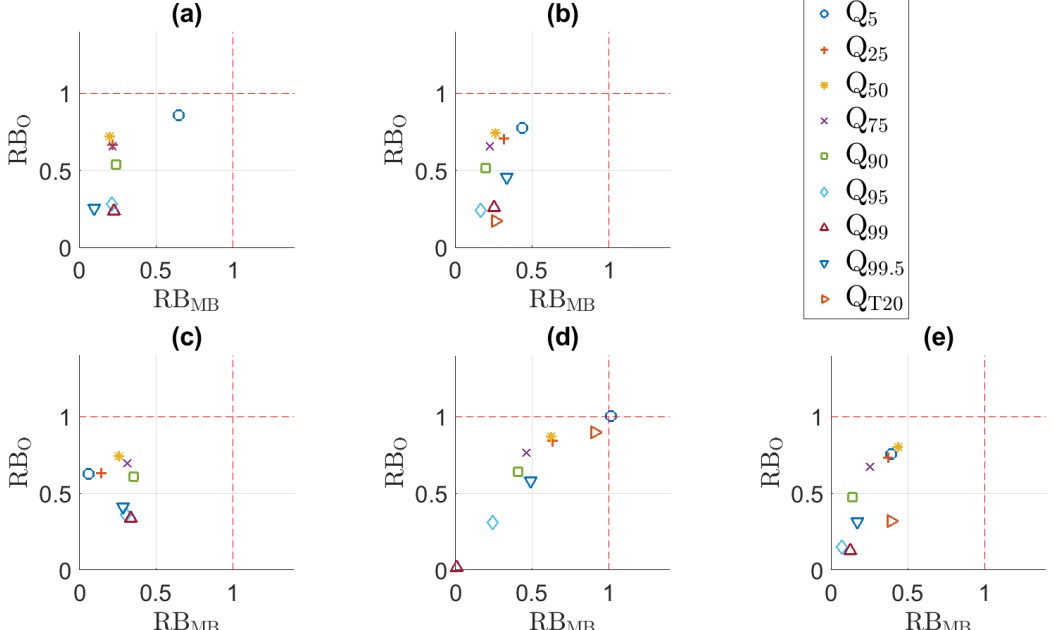

**Figure 10.** $RB_{MB}$ versus $RB_0$ for the discharge percentiles and the 20-year return period value, calculated with raw evaporation. (a) QDM, (b) mQDM, (c) MBCn, (d) MRQNBC, (e) dOTC.

## 6 Conclusions

The goal of this paper was to assess how five bias-adjusting methods handle a climate change context with possible bias nonstationarity. Three of the methods were multivariate bias-adjusting methods: MRQNBC, MBCn and dOTC. The two other

the bias-adjusting methods were univariate: one was a traditional bias-adjusting method, while the other was almost the same method, but modified according to the delta change paradigm. These univariate methods were used as a baseline to compare the multivariate bias-adjusting methods with. The climate change context, using 1970-1989 as calibration time period and 1998-2017 as validation time period, allowed us to calculate the change in bias between the periods, or the extent of bias stationarity, using the R index. All methods were calculated and compared using different indices, for which the residual biases relative to

the observations and model bias were calculated.

The calculated R index values differed depending on the variable and variable index under consideration, but generally demonstrated that the bias of some of these indices is not stationary under climate change conditions. These changes could in some cases be clearly linked to the poor performance of bias-adjusting methods, such as for the lower percentiles of T or the middle percentiles of E. The performance was often poorer for the multivariate bias-adjusting methods, which corroborates the

conclusions of Guo et al. (2020) that bias nonstationarity influences the performance of multivariate bias-adjusting methods. Although these methods have been developed during the last few years as a means to better adjust the biases, it seems that their



more complex calculations make them more vulnerable to bias nonstationarity. Thus, the univariate bias-adjusting methods, computationally less complex and not taking (potentially changing) correlations into account, seem to be more robust. Although effective difference in climate change impact is weakened by the hydrological model we used, the univariate methods still

perform best. Studying other types of climate change impacts, the effect of bias nonstationarity could possible be even larger than discussed here.

The validation results could only be obtained by analysing and comparing a broad combination of indices. Considering only the mean or other standard statistics would have hidden many of the results seen. For example, in contrast to the results for the mean, the inclusion of both high and low extremes highlighted some problems with bias nonstationarity for some variables. As

such, this study does not contradict earlier studies such as Maraun (2012), where the mean-based biases were found to be rather stable. Even a broader set of indices, such as the ETCCDI indices, was not enough to clearly discern between the methods. As such, we repeat the advice by Maraun and Widmann (2018a) to use indices not directly affected by bias-adjusting methods and to analyse the user needs before deciding upon the bias adjustment validation method. An important limitation is that we only used one GCM-RCM-combination. Using a model ensemble will be more informative, but could hide a single model's poor

performance. On the other hand, similar assessments could also be used to discard poor-performing models (expanding upon methods such as those used in e.g. Brunner et al. (2019) or Tokarska et al. (2020)), based on the R index (also suggested by Maurer et al. (2013)) or the remaining bias after adjustment.

The results for the multivariate bias-adjusting methods assessed here are in line with François et al. (2020), especially for the problematic adjustment of occurrence. François et al. (2020) consequently state that the different multivariate bias-adjusting

methods are based on different assumptions, and thus, the end user should make well-grounded choices on the method used. This also became clear in our assessment. However, François et al. (2020) did not study the effect of climate change and bias (non)stationarity and instead focused on model trend preservation, or trend nonstationarity. The results presented and discussed here, such as the contrasting results of MRQNBC and dOTC, imply that whether trend preservation was the focus of a method or not, can have an influence on the bias adjustment. However, it is yet unclear how trend nonstationarity and bias

nonstationarity influence each other and how the most appropriate methods can be discerned, although it has been suggested to use trend-preserving methods whenever we can assume the models to correctly simulate the atmospheric processes (Maraun, 2016).

Although critical of their use, the results of this paper do not imply that multivariate bias-adjusting methods are not helpful. Many of the methods developed during the past few years can also be used for spatial bias adjustment, in which case the loca-

tions can be used as extra variables (see e.g. Vrac (2018)). A similar set-up has not been tested here, but the study by François et al. (2020) has proven the multivariate bias-adjusting methods to be very informative and robust for spatial adjustment: the spatial characteristics that influence local weather the most, such as orography.

Nonetheless, the results discussed in this paper indicate that many methods, and especially the multivariate bias-adjusting methods, fail in handling climate change and its resulting bias nonstationarity correctly. As authors have mentioned before

(Ehret et al., 2012; Maraun, 2016; Nahar et al., 2017), this foremost implies that climate models have to become better at modelling the future: we need to be able to trust them as fully as possible. As long as this is not the case, bias adjustment





methods have to be developed that are more robust and that are able to help us assessing the future correctly. Yet, impact assessment cannot wait for new methods to be developed and/or tested: we need to prepare ourselves for the future as soon as possible. As was shown here, we can state for the current generation of methods that the fewer assumptions and calculations a
method needs, the more robust it is when used in a climate change context. Given this statement, we advise to use univariate bias-adjusting methods, until it becomes more clear how it can be ensured that multivariate methods certainly perform well in a climate change context.

*Code and data availability.* The code for the computations is publicly available at https://doi.org/10.5281/zenodo.4247518 (Hydro-Climate Extremes Lab – Ghent University, 2020). The RCA4 data are downloaded and are available from the Earth System Grid Federation data
repository. The local observations were obtained from RMI in Belgium, and cannot be shared with third parties.

*Author contributions.* JVDV, BDB and NV designed the experiments. JVDV developed the code and performed the calculations. JVDV prepared the manuscript with contributions from MD, BDB and NV. All co-authors contributed to the interpretation of the results.

*Competing interests.* The authors declare that they have no conflict of interests.

*Acknowledgements.* J. Van de Velde would like to thank Y. Robin and R. Mehrotra for some helpful discussion on the use of respectively
dOTC and MRQNBC. The authors are grateful to the RMI for allowing the use of 117-year Uccle dataset. This work was funded by FWO, grant number G.0039.18N.





## Appendix A: Appendix A

Table A1: Observed values, and biases for the raw and adjusted climate simulations.

| Index | Observed value | Bias | | | | | |
|---|---|---|---|---|---|---|---|
| | | Raw climate simulations | QDM | mQDM | MBCn | MRQNBC | dOTC |
| $P_5$ (mm) | 0.00 | 0.00 | 0.00 | 0.00 | 0.00 | -0.40 | 0 |
| $P_{25}$ (mm) | 0.00 | 0.08 | 0.00 | 0.00 | 0.00 | 0.00 | 0.42 |
| $P_{50}$ (mm) | 0.10 | 1.01 | 0.05 | 0.10 | 0.05 | 0.27 | 0.83 |
| $P_{75}$ (mm) | 2.70 | 1.83 | -0.18 | -0.17 | -0.18 | -0.84 | 0.76 |
| $P_{90}$ (mm) | 7.40 | 1.99 | -0.26 | -0.30 | -0.26 | -1.36 | -0.23 |
| $P_{95}$ (mm) | 11.42 | 2.38 | -0.61 | -0.60 | -0.61 | -1.44 | -0.65 |
| $P_{99}$ (mm) | 21.80 | 2.36 | -1.86 | -1.73 | -1.86 | 1.38 | -1.55 |
| $P_{99.5}$ (mm) | 29.09 | 1.56 | -4.20 | -3.97 | -4.20 | 0.02 | -4.02 |
| $T_5$ (ºC) | 0.40 | -0.31 | -0.70 | -1.43 | -0.70 | -0.63 | -0.94 |
| $T_{25}$ (ºC) | 6.30 | -0.08 | -0.68 | -1.55 | -0.68 | -3.00 | -0.73 |
| $T_{50}$ (ºC) | 11.40 | -0.40 | -0.81 | -0.88 | -0.81 | -3.24 | -0.70 |
| $T_{75}$ (ºC) | 16.10 | -0.70 | -0.46 | 0.09 | -0.46 | -1.07 | -0.37 |
| $T_{90}$ (ºC) | 19.40 | -1.07 | -0.35 | 0.31 | -0.35 | 1.02 | 0.12 |
| $T_{95}$ (ºC) | 21.30 | -1.17 | -0.01 | 0.37 | -0.01 | 2.59 | 0.25 |
| $T_{99}$ (ºC) | 24.95 | -1.85 | -0.16 | -0.75 | -0.16 | 5.95 | 0.46 |
| $T_{99.5}$ (ºC) | 25.90 | -1.80 | 0.16 | -1.39 | 0.16 | 7.01 | 0.77 |
| $E_5$ (mm) | 0.00 | 0.20 | 0.00 | 0.00 | 0.00 | -0.04 | 0 |
| $E_{25}$ (mm) | 0.52 | 0.15 | -0.09 | -0.10 | -0.09 | -0.16 | -0.52 |
| $E_{50}$ (mm) | 1.42 | 0.05 | -0.27 | -0.28 | -0.27 | -0.38 | -0.77 |
| $E_{75}$ (mm) | 2.69 | -0.02 | -0.34 | -0.35 | -0.34 | -0.58 | -0.65 |
| $E_{90}$ (mm) | 3.65 | 0.10 | -0.27 | -0.27 | -0.27 | -0.47 | -0.18 |
| $E_{95}$ (mm) | 4.21 | 0.15 | -0.30 | -0.28 | -0.30 | -0.41 | 0.125 |
| $E_{99}$ (mm) | 5.02 | 0.21 | -0.16 | -0.13 | -0.16 | -0.17 | 0.69 |
| $E_{99.5}$ (mm) | 5.24 | 0.27 | -0.10 | -0.02 | -0.10 | -0.07 | 1.03 |
| $corr_{P,E}$ (-) | -0.18 | -0.04 | -0.06 | 0.02 | 0.19 | 0.17 | 0.57 |
| $corr_{P,T}$ (-) | -0.16 | 0.18 | 0.14 | 0.09 | 0.16 | 0.16 | 0.04 |
| $corr_{E,T}$ (-) | 0.82 | -0.02 | -0.03 | -0.09 | -0.84 | 0.02 | -0.45 |





| | | | | | | | |
|---|---|---|---|---|---|---|---|
| crosscorr$_{P,E,0}$ (-) | 0.30 | 0.06 | -0.04 | -0.02 | 0.05 | -0.03 | 0.12 |
| crosscorr$_{P,E,1}$ (-) | 0.24 | 0.19 | 0.08 | -0.01 | 0.10 | 0.05 | 0.11 |
| crosscorr$_{P,T,0}$ (-) | 0.36 | 0.15 | 0.05 | 0.01 | 0.02 | -0.03 | 0.06 |
| crosscorr$_{P,T,1}$ (-) | 0.38 | 0.13 | 0.02 | -0.01 | 0.02 | -0.04 | 0.08 |
| crosscorr$_{E,T,0}$ (-) | 0.93 | -0.00 | -0.02 | -0.05 | -0.29 | -0.02 | -0.21 |
| crosscorr$_{E,T,1}$ (-) | 0.91 | 0.01 | -0.01 | -0.04 | -0.27 | -0.01 | -0.24 |
| $P_{P00}$ (-) | 0.65 | -0.10 | -0.00 | -0.02 | -0.17 | 0.00 | -0.37 |
| $P_{P10}$ (-) | 0.32 | -0.15 | 0.00 | -0.05 | 0.16 | -0.13 | -0.07 |
| $N_{dry}$ (-) | 3470.00 | -1466.00 | 0.00 | -373.00 | 0.00 | -923.00 | -1604.20 |
| $P_{lag1}$ (-) | 0.33 | 0.11 | 0.02 | 0.07 | -0.12 | 0.08 | 0.05 |
| $Q_5$ (m$^3$/s) | 2.30 | 0.92 | -0.32 | -0.18 | 0.82 | -0.40 | 1.50 |
| $Q_{25}$ (m$^3$/s) | 3.36 | 1.45 | 0.02 | -0.12 | 0.29 | -0.23 | 1.50 |
| $Q_{50}$ (m$^3$/s) | 4.39 | 1.53 | 0.08 | 0.02 | -0.20 | -0.42 | 1.33 |
| $Q_{75}$ (m$^3$/s) | 5.72 | 2.52 | -0.08 | -0.03 | -0.72 | -0.81 | 1.51 |
| $Q_{90}$ (m$^3$/s) | 7.83 | 4.76 | -0.36 | -0.07 | -1.66 | -1.37 | 2.12 |
| $Q_{95}$ (m$^3$/s) | 10.09 | 9.22 | -1.00 | -0.33 | -2.78 | -1.78 | 2.94 |
| $Q_{99}$ (m$^3$/s) | 18.71 | 18.58 | -1.65 | -0.78 | -3.21 | 4.77 | 5.77 |
| $Q_{99.5}$ (m$^3$/s) | 23.90 | 19.70 | 0.84 | -1.77 | -0.57 | 13.81 | 6.61 |
| $Q_{T20}$ (m$^3$/s) | 48.69 | 54.61 | 8.36 | -3.41 | -10.40 | 52.45 | 25.03 |





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
