# Peer review of "Impact of bias nonstationarity on the performance of uni- and multivariate bias-adjusting methods: a case study on data from Uccle, Belgium"

_Hydrology and Earth System Sciences, 2020_

## Referee Comment (RC1) · Bastien FRANCOIS (Referee) · 11 Jan 2021

The authors of "Impact of bias nonstationarity on the performance of uni- and multivariate bias-adjusting methods" apply different methods of bias correction (BC) for the adjustment of temperature, precipitation and evaporation time series. Then, adjusted precipitation and evaporation time series are given to a hydrological model for impact assessment. In particular, the performance of 2 univariate and 3 multivariate BC methods are assessed under climate change conditions in order to determine the influence of bias nonstationarity on results. The authors find that the multivariate BC (MBC) methods often perform worse than the univariate BC (1d-BC) methods concerning the

adjustment of the marginal and inter-variable properties of the variables for the projection period. These underperformances from MBCs for the correction of the selected simulated variables are linked with their changes under climate change conditions, suggesting that MBCs are failing to cope with nonstationary climate conditions.

While I see the interest in assessing the influence of bias nonstationarity on results from BC methods, I am not sure that the present study is comprehensive enough to do so precisely and to provide appropriate nuances. This however can be improved by several means.

General comments:

1. Results are often not well explained when bad performances for (M)BCs are obtained. The validation metrics used in this study (residual biases relative to the observations (Rbo) and to the model bias (Rbmb)) tend to make, in my opinion, the analyses of results difficult for readers. In that respect, I would find helpful to provide raw values of the indices for, respectively, the calibration and projection periods (i.e. without computing directly their biases as in Table A1) and their changes between the calibration and projection periods for: the reference, the simulations and the 5 corrections from the (M)BC methods. In a general way, it would permit to better explain the results obtained from the (M)BC methods, i.e. whether good or bad results for a specific method come from the nonstationary problem (i.e. from the fact that simulated changes are "wrong", which alter the quality of results from MBC methods) or from the characteristics of the method (e.g., the statistical technique used, its stochastic nature, etc.). Identifying the reasons for good or bad results from (M)BC methods is of key importance for this study, to be able to conclude with more certainty that bias nonstationarity is the main source of problem.

2. The authors conclude that "the univariate bias-adjusting methods, computationally less complex and not taking (potentially changing) correlations into account, seem to be more robust." (L787). Concerning correlation, I am afraid that this result of robust-

ness is only specific to the present application, and hence the generalization of this conclusion cannot be done as the authors do (e.g., L825 "we advise to use univariate bias-adjusting methods, until it becomes more clear how it can be ensured that multi-variate methods certainly perform well in a climate change context."). Indeed, one of the advantages of considering MBC methods instead of univariate ones is that MBCs are able to adjust correlations between variables. Univariate BC methods are not designed to do so. For example, 1D-BC methods such as QDM globally conserve the rank structures of the climate model to correct, and hence the simulated dependence structure between variables is preserved. According to Table A1, simulated correlations often present little bias compared to observations (for example, for corr(P,E), corr(E,T), crosscorr(P,E,0), crosscorr(E,T,0), crosscorr(E,T,1)). Then, by preserving the simulated rank structures, 1d-BC methods like QDM mechanically present correlations with little bias as well. This result is really specific to this study, and would not be obtained if the raw climate simulations would have presented strong biases in correlations. Hence, concerning correlations, results from 1d-BC as QDM depend on how well the models simulate relevant dependencies between the climate variables. This is something already suggested by Zscheischler et al. (2019). This point is not explained in the present study, while it is one of the key points of discussion, and the principal reason why QDM often performs well in this study for correlations. This point should be mentioned and discussed to provide the appropriate nuances to initial conclusions.

3. Linked with the comment 1. above, I would find interesting to provide (at least in Appendix) the results for the multivariate bias correction and the hydrological model for the calibration period. In my opinion, verifying that MBC methods perform well for the calibration period (which has to be verified) would validate the global methodology and would better support the conclusions on the effect of bias nonstationarity on the results from MBCs for future periods.

4. As explained by the authors (L79), climate models can present statistical biases compared to observations that are nonstationary, that is, that the differences of bias

between the calibration and future period are not the same. What is often not clarified clearly through the article is that, in a changing climate context, another way to see bias nonstationarity is that it results from the fact that observed and simulated variables do not present the same changes between the calibration and projection periods. This is of key importance, as some of the (M)BC methods included in this study can take into account the simulated changes in their correction procedures (such as dOTC or MBCn). It must be better highlighted through the article, and has to be used to provide a better analysis of the results. 5. I think the reasons why the three MBCs (MRQNBC, dOTC and MBCn) are selected in this study must be better specified. In particular, their differences of assumptions and attributes could be better indicated, for example with a table (as recommended in 6.).

6. If I understood well, all the three MBC methods are able to take into account (some of) the potential simulated changes in their correction procedures. It also exists in the literature MBC methods that assume dependence structure to be stable in time (R2D2, Vrac, 2018, Vrac and Thao, 2020). It would have been interesting to include in the study such MBC methods as benchmarks for a better assessment of the potential losses (or benefits) of considering stability of dependence structure, even in a changing climate context.

7. The quality of presentation should be improved.

- The "Bias-adjusting methods" section is too long. Please consider to summarize each of the BC methods and their main properties in a short text (and with the help of a table, as already explained) to keep from overwhelming the reader with technical details that are not necessarily useful to get the main points. Technical details and algorithms can eventually be placed in Appendix.

- Please consider to indicate the letters for each figure when describing the results (for instance, for QDM: Figure 3a, mQDM: Figure 3b, etc). This would better guide the reader through the different plots.

Specific comments:

L55: I would replace the word "uncertainty" as it can be misleading here. I would rather say "error (or bias) that can propagate in the impact models".

L72: in François et al. (2020), it is explained that all multivariate methods that are under study fail in adjusting the temporal structure of simulated times series. Of course, it already exists methods that can adjust (part of) the temporal properties: MRQNBC is one of them.

L75: what do you mean by "more recent validation period". I think that just mentioning "validation period" is enough.

L225: Did you verify that assuming the stationarity of the frequency of dry days holds for your application? Also, did you apply a thresholding after bias correction? I know that, for example, dOTC can generate negative values for precipitation. How did you deal with this problem?

L282: The successive conditional approach performs successive corrections conditionally on the variables already corrected. This can be applied to more than two variables. It should be specified. It does not "adjust a second variable conditionally on the second variable", as written.

L286: This is not well explained. The problem of robustness of the successive conditional approach is specific for a high number of dimensions to correct. Indeed, in a successive conditional approach, as the number of variables already corrected increases at each successive step, it progressively reduces the number of data available for the correction, making it less and less robust.

L396: Do you know why the results are exacerbated?

L429-440: I am not sure that the equations to explain the Schaake Shuffle are necessary, as it can simply be explained with words.

L442: Do you have a reference to account for ties by introducing small random values? There exists also other ways to compute ranks to handle problems of equal values.

L445: This result is really "case-specific". I would precise it, as it cannot be generalized for each application (depending on the bias-nonstationarity, for example).

L454: Did you check if, even after early stopping, overfitting was not a problemÂă? In particular, how did you choose the tolerance of 0.0001?

L463: I would precise that dOTC extends the CDFt method to the multivariate case.

L465: dOTC is indeed designed to preserve the trend of the model for marginal properties but also for dependence structure (or copula). It should be specified, as it is a major difference with univariate methods.

L520: I don't think that the 10 calculations made for dOTC are necessary. What are the bin sizes chosen to implement dOTC? If the bin sizes are small, the influence of the stochastic components in dOTC is rather small, and hence introducing 10 calculations does not seem necessary to me.

L593: I do not understand the sentence "Yet, QDM has the best RBMB values and might thus be preferable", that seems in opposition with the previous sentence.

L644: "an influence" on what? Correlations? Please be more precise.

L677: Actually, it surprises me as it has been found in François et al. (2020) that, for a small number of dimensions to correct, MBCn and dOTC preserve roughly the rank sequences of the model to correct. It normally does not have such "important shuffling" as specified. Do you know why?

L698: A difference between QDM and MBCn is also the adjustment of dependence structure. It should be specified.

L705-706: I do not understand. Please rewrite.

L790: Did you mean "possibly"?

Bibliography

Zscheischler, J., Fischer, E. M., and Lange, S.: The effect of univariate bias adjustment on multivariate hazard estimates, Earth System Dynamics, 10, 31–43, https://doi.org/10.5194/esd-10-31-2019, 2019.

Vrac, M.: Multivariate bias adjustment of high-dimensional climate simulations: the Rank Resampling for Distributions and Dependences (R2D2) bias correction, Hydrology and Earth System Sciences, 22, 3175, https://doi.org/10.5194/hess-22-3175-2018, 2018.

Vrac, M. and Thao, S.: R2D2 v2.0: Accounting for temporal dependences in multivariate bias correction via analogue ranks resampling, Geosci. Model Dev., 2020, 1–29, https://doi.org/10.5194/gmd-2020-132, 2020.

François, B., Vrac, M., Cannon, A. J., Robin, Y., and Allard, D.: Multivariate bias corrections of climate simulations: Which benefits for which losses?, Earth Syst. Dyn., 2020, 1–41, https://doi.org/10.5194/esd-2020-10, 2020.

---

## Referee Comment (RC2) · Anonymous Referee #2 · 29 Jan 2021

This manuscript evaluated the impact of bias nonstationarity on the performance of uni- and multivariate bias correction methods. Specifically, five bias correction methods (including one univariate bias correction method QDM, one delta change method mQDM, and three multivariate methods MBCn, MRQNBC, and dOTC) were compared in terms of correcting the precipitation, potential evaporation, and temperature. The authors concluded that these newly proposed multivariate bias correction methods may perform worse than the commonly used univariate method in both climate and hydrology perspectives, due to the bias nonstationarity of climate model simulations. The topic is very interesting and is also important in climate and hydrology communities. However, I found the manuscript is hard to read, especially the experiment design and

the conclusion may be unreliable. My major comments are as follows

1. The authors did not explicitly present the study area of this paper. After looking for a while, I'm surprised to find that only one station and one grid cell were used (lines 115-143). In addition, there is only one GCM-RCM combination was used, and no information on its spatial resolution. As such, the results of this study are subjected to large uncertainty. 2. The manuscript is lengthy and hard to follow. I think some information does not need to be presented in detailed in the manuscript. For example, the introduction of the bias correction methods (section 3.1 to 3.3, almost 14 pages of the main body). All these information can be found in literatures. 3. Although the authors have stated in line 527 "As the effect on discharge is the overarching goal of this paper", I think the information on the hydrological model and hydrological simulation this quite poor in the manuscript. Firstly, I did not find how the hydrological model performed in the study area (e.g. Nash value or some other criteria). Although the goal of this study is to compare the difference between the univariate method and multivariate method in the hydrology perspective, the hydrological model should be well calibrated. Secondly, as is stated that the PDM model was not calibrated in Uccle but Grote Nete watershed, please give the evidence to show that it is feasible to drive the PDM using the climate data in Uccle. 4. The authors used scatter plots throughout the whole paper. It is difficult to see how many dots are located within the 0-1 square or to do the comparison (e.g. Figure 6). I suggest that maybe the authors can use some more quantitative metrics and figures to show the results. 5. Figure 3 to figure 9, the authors did not mention which period (calibration or validation) and which month the results are based, or is it based on the whole year? In this case, readers cannot understand and assess the results. For example, for the correlations between two variables (e.g. P and T), the correlation coefficients are quite different for each month (e.g. summer and winter), therefore, at least, they should be evaluated separately for each month. 6. Table 1, I am wondering whether the Spearman correlation coefficients and lag-0 cross-correlation between two variables reflect the same thing. 7. In general, I found that many expressions in the current manuscript are not very accurate. For

example, in line 568 the authors write "A surprising result for P is the high RBMB value for P99.5 for MRQNBC", but I found in figure 3 that the corresponding value for MRQNBC is quite small. Therefore, I'm quite confused by many expressions in the current manuscript.

———————————————

---

## Author Comment (AC1) · 10 Mar 2021

**Response to Referee Comment 1 on 'Impact of bias nonstationarity on the performance of uni- and multivariate bias-adjusting methods' by J. Van de Velde et al.**

**Bastien François (Referee)**

We would like to thank Bastien François for his fair and thorough review. Below, we give a comment-by-comment response, indicating the changes we plan to make to the manuscript.

**General Comments**

Comment: Results are often not well explained when bad performances for (M)BCs are obtained. The validation metrics used in this study (residual biases relative to the observations (Rbo) and to the model bias (Rbmb)) tend to make, in my opinion, the analyses of results difficult for readers. In that respect, I would find helpful to provide raw values of the indices for, respectively, the calibration and projection periods (i.e. without computing directly their biases as in Table A1) and their changes between the calibration and projection periods for: the reference, the simulations and the 5 corrections from the (M)BC methods. In a general way, it would permit to better explain the results obtained from the (M)BC methods, i.e. whether good or bad results for a specific method come from the nonstationary problem (i.e. from the fact that simulated changes are "wrong", which alter the quality of results from MBC methods) or from the characteristics of the method (e.g., the statistical technique used, its stochastic nature, etc.). Identifying the reasons for good or bad results from (M)BC methods is of key importance for this study, to be able to conclude with more certainty that bias nonstationarity is the main source of problem.

**Response:** We used the RBo and RBmb metrics as a method to present detailed information on the full extent of changes in a visual way and to enable the reader an easier comparison without needing too many tables. However, these are new metrics and we agree that the raw values are a valuable source of additional information that can provide additional guidance to the reader. However, as referee #2 preferred seasonal or monthly data, providing all raw values and biases would seriously lengthen the paper. As the RBo and RBmb values can help understand the bias adjustment behavior at some points, we prefer to keep those in the paper. Nonetheless, we will add the original values whenever necessary for interpreting the results and at least provide all raw values in a supplement to the paper.

Comment: The authors conclude that "the univariate bias-adjusting methods, computationally less complex and not taking (potentially changing) correlations into account, seem to be more robust." (L787). Concerning correlation, I am afraid that this result of robust-ness is only specific to the present application, and hence the generalization of this conclusion cannot be done as the authors do (e.g., L825 "we advise to use univariate bias-adjusting methods, until it becomes more clear how it can be ensured that multivariate methods certainly perform well in a climate change context."). Indeed, one of the advantages of considering MBC methods instead of univariate ones is that MBCs are able to adjust correlations between variables. Univariate BC methods are not designed to do so. For example, 1D-BC methods such as QDM globally conserve the rank structures of the climate model to correct, and hence the simulated dependence structure between variables is preserved. According to Table A1, simulated correlations often present little bias compared to observations (for example, for corr(P,E), corr(E,T), crosscorr(P,E,0), crosscorr(E,T,0), crosscorr(E,T,1)). Then, by preserving the simulated rank structures, 1d-BC methods like QDM mechanically present correlations with little bias as well. This result is really specific to this study, and would not be obtained if the raw climate simulations would have presented strong biases in correlations. Hence, concerning correlations, results from 1d-BC as QDM depend on how well the models simulate relevant dependencies between the climate variables. This is something already suggested by Zscheischler et al. (2019). This point is not explained in the present study, while it is one of the key points of discussion, and the principal reason why QDM often performs well in this study for

correlations. This point should be mentioned and discussed to provide the appropriate nuances to initial conclusions.

**Response:** The bias in correlation and the simulation of dependencies is a point that should have been mentioned better. We will address this in the revised manuscript in the section on correlation results and the discussion.

Comment: Linked with the comment 1. above, I would find interesting to provide (at least in Appendix) the results for the multivariate bias correction and the hydrological model for the calibration period. In my opinion, verifying that MBC methods perform well for the calibration period (which has to be verified) would validate the global methodology and would better support the conclusions on the effect of bias nonstationarity on the results from MBCs for future periods.

**Response:** We have calculated the adjustment for all methods in the calibration period. The information will be included in the revised version. This shows, in combination with the seasonal results requested by referee #2 that the original statement that multivariate bias-adjusting methods perform poorer than the univariate bias-adjusting methods was an overstatement. Nonetheless, the comparison illustrates that 1) for some seasons, there is a clear influence of bias nonstationarity on the results and 2) that the multivariate bias-adjusting methods are less well equipped to deal with the seasonal differences in bias nonstationarity.

Comment: As explained by the authors (L79), climate models can present statistical biases compared to observations that are nonstationary, that is, that the differences of bias between the calibration and future period are not the same. What is often not clarified clearly through the article is that, in a changing climate context, another way to see bias nonstationarity is that it results from the fact that observed and simulated variables do not present the same changes between the calibration and projection periods. This is of key importance, as some of the (M)BC methods included in this study can take into account the simulated changes in their correction procedures (such as dOTC or MBCn). It must be better highlighted through the article, and has to be used to provide a better analysis of the results.

**Response:** This information will be added to the introduction of Section 3.3 and considered in the Results, Discussions and Conclusions sections where necessary.

Comment: I think the reasons why the three MBCs (MRQNBC, dOTC and MBCn) are selected in this study must be better specified. In particular, their differences of assumptions and attributes could be better indicated, for example with a table (as recommended in 6.).

**Response:** The introduction for the multivariate bias-adjusting methods will be extended and an overview table will be added to the end of this discussion. With this table, the reader should be able to get a clearer overview of the differences between the methods.

Comment: If I understood well, all the three MBC methods are able to take into account (some of) the potential simulated changes in their correction procedures. It also exists in the literature MBC methods that assume dependence structure to be stable in time (R2D2, Vrac, 2018, Vrac and Thao, 2020). It would have been interesting to include in the study such MBC methods as benchmarks for a better assessment of the potential losses (or benefits) of considering stability of dependence structure, even in a changing climate context.

**Response:** The R2D2 method is indeed a valuable method to include in the analysis. We have already computed the adjustment by this method and will include the method. This also implies that Section 3 and the detailed information in the Appendix will be extended to give a good overview of this method.

Comment: The quality of presentation should be improved.
- The "Bias-adjusting methods" section is too long. Please consider to summarize each of the BC methods and their main properties in a short text (and with the help of a table, as already explained)

to keep from overwhelming the reader with technical details that are not necessarily useful to get the main points. Technical details and algorithms can eventually be placed in Appendix.
- Please consider to indicate the letters for each figure when describing the results (for instance, for QDM: Figure 3a, mQDM: Figure 3b, etc). This would better guide the reader through the different plots.

**Response:** As suggested, the technical details of the methods will be moved to the Appendix. As mentioned previously, a table will be added in the subsection on multivariate methods to better guide the reader. Where needed, the figure panel will be specified to better guide the readers.

**Specific Comments**

Comment: L55: I would replace the word "uncertainty" as it can be misleading here. I would rather say "error (or bias) that can propagate in the impact models".

**Response:** Thank you for the suggestions, this will be updated.

Comment: L72: in François et al. (2020), it is explained that all multivariate methods that are under study fail in adjusting the temporal structure of simulated times series. Of course, it already exists methods that can adjust (part of) the temporal properties: MRQNBC is one of them.

**Response:** The sentence will be improved to better reflect this nuance.

*Besides, they also noticed that all multivariate methods studied fail in adjusting the temporal structure of a time series.*

Comment: L75: what do you mean by "more recent validation period". I think that just mentioning "validation period" is enough.

**Response:** This will be updated in the paper.

Comment: L225: Did you verify that assuming the stationarity of the frequency of dry days holds for your application? Also, did you apply a thresholding after bias correction? I know that, for example, dOTC can generate negative values for precipitation. How did you deal with this problem?

**Response:** As is shown in the results (R index values) the number of dry days can be considered stationary for the time periods studied. A reference to the first Section of the results will be added. For dOTC, we applied a thresholding before calculating the indices, as this was suggested in Robin et al. (2019). However, this was not mentioned, and will be added to the discussion of dOTC in the Appendix.

(Number of dry days) *However, as thresholding is used prior to all methods, the influence of possible bias nonstationarity on ∆N is assumed to be negligible. Besides, as is shown in Section 4.1 the number of dry days is stationary for the time frames studied in this paper.*

(dOTC) *Robin et al. (2019) warn that this final result could contain negative values for precipitation and thus a thresholding has to be applied before the data can be used. This thresholding consists of setting all negative precipitation values to zero.*

Comment: L282: The successive conditional approach performs successive corrections conditionally on the variables already corrected. This can be applied to more than two variables. It should be specified. It does not "adjust a second variable conditionally on the second variable", as written.

**Response:** This sentence was indeed incorrect. We will clarify that it can be applied to more than two variables.

*These two components are then recombined to obtain data that are close to the observations for both marginal and multivariate aspects. The latter approach consists of adjusting a variable conditionally on the variables already adjusted. This procedure is applied successively to each variable.*

Comment: L286: This is not well explained. The problem of robustness of the successive conditional approach is specific for a high number of dimensions to correct. Indeed, in a successive conditional approach, as the number of variables already corrected increases at each successive step, it progressively reduces the number of data available for the correction, making it less and less robust.

**Response:** This will be better clarified in the paper.

*According to Vrac (2018) the latter approach suffers from two main limitations. First, the adjustment is performed conditionally on the previously adjusted data, which decreases the robustness of each successive step. At each of these steps, the number of variables already adjusted increases and hence the amount of data available for adjustment decreases, reducing the robustness.*

Comment: L396: Do you know why the results are exacerbated?

**Response:** The first repetition created some unphysical values (especially for precipitation). As the implementation is based on the correlations, every repetition exacerbated these nonphysical values. However, when running the method just once, the results were satisfactory. This will be updated in the paper. The method is not entirely suited for bounded variables such as precipitation and evaporation. Nonetheless, as some of the results were good in both calibration and validation periods, methods based on the same principles are worth investigating and using for precipitation.

*However, the nesting method cannot fully remove biases at all time scales, thus Mehrotra et al. (2016) suggested to repeat the complete procedure multiple times. Yet, in our case multiple repetitions exacerbated the results. Nonphysical outliers created by the first repetition influenced the subsequent repetitions, creating even more nonphysical values. This was most clearly seen for precipitation. As a bounded variable, precipitation is most sensitive for nonphysical values. Nonetheless, running the method just once yielded agreeable results.*

Comment: L429-440: I am not sure that the equations to explain the Schaake Shuffle are necessary, as it can simply be explained with words.

**Response:** Not every written explanation of the Schaake Shuffle is equally clear, and we wanted to ensure that readers could follow by providing an exact mathematical formulation. Although it is true that this burdens the reader with additional equations, these will be moved to the appendices along with the other details on the methods.

Comment: L442: Do you have a reference to account for ties by introducing small random values? There exists also other ways to compute ranks to handle problems of equal values.

**Response:** This was based on the approach used in Vandenberghe et al. (2010), in which ties were removed before copulas were fit. Vandenberghe et al. (2010) based this on the discussions in Salvadori and De Michele (2006) and Salvadori et al. (2007). These references will be added in the paper.

Comment: L445: This result is really "case-specific". I would precise it, as it cannot be generalized for each application (depending on the bias-nonstationarity, for example).

**Response**: This will be rephrased to better accentuate the case-specificity.

*In the study of Cannon (2018), MBCn outperformed many earlier multivariate bias-adjusting methods, such as the EC-BC, the JBC, MBCr and MBCp methods. However, …*

Comment: L454: Did you check if, even after early stopping, overfitting was not a problem? In particular, how did you choose the tolerance of 0.0001?

Response: We checked overfitting by studying the final distance scores: these show that there is still some margin for improvement. The tolerance was chosen by trial and error: this level of tolerance yielded fairly good results while not increasing the duration too much. Will be added the text.

*A tolerance of 0.0001 was used: if the difference between two consecutively calculated energy distances was lower, the computation was halted. The tolerance was chosen on the basis of trial and error: the final value provided a good balance of duration and bias adjustment.*

Comment: L463: I would precise that dOTC extends the CDFt method to the multivariate case.

**Response:** This will be added to the paper.

Comment: L465: dOTC is indeed designed to preserve the trend of the model for marginal properties but also for dependence structure (or copula). It should be specified, as it is a major difference with univariate methods.

**Response:** This specification is indeed important and will be added to the paper.

*The combination of both optimal transport plans allows for bias adjustment while preserving the trend of the model for both marginal properties and the dependence structure.*

Comment: L520: I don't think that the 10 calculations made for dOTC are necessary. What are the bin sizes chosen to implement dOTC? If the bin sizes are small, the influence of the stochastic components in dOTC is rather small, and hence introducing 10 calculations does not seem necessary to me.

**Response:** For each variable, 25 bins were used. More bins quickly increased the computational load. The bin size depends on the variable space. The results differed among the calculations, and thus it is better to keep them and average the results. A note on the influence of the stochastic components will be added to the detailed overview of the method in the Appendix.

*This ensemble accounts for random weather effects and can thus be considered to be more similar to the true range of observations. However, the true stochasticity depends on the cell size. By decreasing the cell size, the influence of the stochastic effects will also decrease. Thus, depending on the cell size, accounting for the stochasticity may or may not be necessary.*

Comment: L593: I do not understand the sentence "Yet, QDM has the best RBMB values and might thus be preferable", that seems in opposition with the previous sentence.

**Response:** The results will be rewritten based on seasonal results (a request of Referee #2) and this sentence has been removed.

Comment: L644: "an influence" on what? Correlations? Please be more precise.

**Response:** The results will be rewritten based on seasonal results (a request of Referee #2) and this sentence will be removed.

Comment: L677: Actually, it surprises me as it has been found in François et al. (2020) that, for a small number of dimensions to correct, MBCn and dOTC preserve roughly the rank sequences of the model to correct. It normally does not have such "important shuffling" as specified. Do you know why?

**Response:** This should have been written differently. As can be seen in Figure 7, the RBmb and RBo values for the lag-1 autocorrelation and wet-dry transition probability (panel (c), MBCn) and the

number of dry days (panel (e), dOTC) are close to 1, indicating that they are very close to the original climate simulation values. Thus, the methods indeed roughly preserve the rank sequence. As the results will be rewritten based on the seasonal results, this sentence will be removed. However, we will ensure that the writing is clearer on this specific aspect of the method.

Comment: L698: A difference between QDM and MBCn is also the adjustment of dependence structure. It should be specified.

**Response:** Thank you for addressing this point. Although the specific sentence will probably be removed, we will take this point into account when revising the Results section.

Comment: L705-706: I do not understand. Please rewrite.

**Response:** The results will be rewritten based on seasonal results (a request of Referee #2) and this sentence will be removed.

Comment: Did you mean "possibly"?

**Response:** Yes, thank you for noting this mistake.

**References**

Salvadori, G. and De Michele, C.: Statistical characterization of temporal structure of storms, Advances in Water Resources, 29, 827–842, https://doi.org/10.1016/j.advwatres.2005.07.013, 2006

Salvadori, G., De Michele, C., Kottegoda, N. T., and Rosso, R.: Extremes in nature: an approach using copulas, vol. 56, Springer Science &Business Media, 2007

Vandenberghe, S., Verhoest, N. E. C., and De Baets, B.: Fitting bivariate copulas to the dependence structure between storm characteristics: A detailed analysis based on 105 year 10 min rainfall, Water resources research, 46, https://doi.org/10.1029/2009WR007857, 2010

---

## Author Comment (AC2) · 10 Mar 2021

**Response to Referee Comment 2 on 'Impact of bias nonstationarity on the performance of uni- and multivariate bias-adjusting methods' by J. Van de Velde et al.**

**Anonymous referee**

We would like to thank the referee for the time spent reviewing our manuscript. Below, we give a comment-by-comment response.

**General Comments**

Comment: The authors did not explicitly present the study area of this paper. After looking for a while, I'm surprised to find that only one station and one grid cell were used (lines 115-143). In addition, there is only one GCM-RCM combination was used, and no information on its spatial resolution. As such, the results of this study are subjected to large uncertainty.

**Response:** We believe our research design is still informative, as it is too date scarcely addressed and can provide a base-line for more extended follow-up studies. Yet we simultaneously agree on the limitation of our design, and will ensure these are properly stated both the Introduction and the Discussion.

Comment: The manuscript is lengthy and hard to follow. I think some information does not need to be presented in detailed in the manuscript. For example, the introduction of the bias correction methods (section 3.1 to 3.3, almost 14 pages of the main body). All these information can be found in literatures.

**Response:** We prefer not to completely remove the technical information, but move it to the Appendix instead, as suggested by Referee #1. The reasons for not removing the information are threefold. First, not every original paper is easy to follow without understanding additional technical details. In our paper, we tried to find a balance between mathematical background and the essence of the paper. Second, we have made some small changes to implementations and these are easier to understand with the full background. Third, the characteristics of the multivariate methods are of great importance to the results and to give the reader an easy reference, we wanted to have the most important information close by.

Comment: Although the authors have stated in line 527 "As the effect on discharge is the overarching goal of this paper", I think the information on the hydrological model and hydrological simulation this quite poor in the manuscript. Firstly, I did not find how the hydrological model performed in the study area (e.g. Nash value or some other criteria). Although the goal of this study is to compare the difference between the univariate method and multivariate method in the hydrology perspective, the hydrological model should be well calibrated. Secondly, as is stated that the PDM model was not calibrated in Uccle but Grote Nete watershed, please give the evidence to show that it is feasible to drive the PDM using the climate data in Uccle.

**Response:** Some details on the calibration performance will be added. Besides, the Grote Nete watershed is roughly 50 km from the Uccle station and the effect of topography on weather is negligible. Hence, the assumption that the Uccle data can be used for the watershed is acceptable, as discussed in the referenced and other earlier papers.

Comment: The authors used scatter plots throughout the whole paper. It is difficult to see how many dots are located within the 0-1 square or to do the comparison (e.g. Figure 6). I suggest that maybe the authors can use some more quantitative metrics and figures to show the results.

**Response:** The scatter plots allowed for the inclusion of multiple indices or interesting percentiles of the variables' distribution in one plot and are hence practical for conveying information. However,

we understand that using only the RBmb and RBo values might impede some readers in quickly grasping the results. As such, we will also include other measures, such as Perkins Skill Score and ETCCDI indices, wherever applicable. Yet, not every metric is as practical. For example, while the RMSE provides general information on the fit, it hides the detailed information on the percentiles. Nonetheless, by balancing both new and standard metrics, we aim to provide a good overview.

Comment: Figure 3 to figure 9, the authors did not mention which period (calibration or validation) and which month the results are based, or is it based on the whole year? In this case, readers cannot understand and assess the results. For example, for the correlations between two variables (e.g. P and T), the correlation coefficients are quite different for each month (e.g. summer and winter), therefore, at least, they should be evaluated separately for each month.

**Response:** These figures were based on the validation period, with data for the whole year. All figures will be updated to clarify the time frame. Besides, as suggested, we have calculated monthly and seasonal data. This indeed allows for a better assessment of the inter-seasonal differences, which provides interesting insights. To provide a balanced overview of the results and keep a proper length of the paper, we prefer the seasonal evaluation. All results will be rewritten to reflect the updated validation.

Comment: Table 1, I am wondering whether the Spearman correlation coefficients and lag-0 cross-correlation between two variables reflect the same thing.

**Response:** The Spearman correlation is based on the rank of the value, the crosscorrelation is based on its effective value; these two values thus do not reflect the same thing. This can also be seen in Figure 7, where the RBmb and RBo values for the Spearman correlation and the always differ by some extent.

Comment: In general, I found that many expressions in the current manuscript are not very accurate. For example, in line 568 the authors write "A surprising result for P is the high RBMB value for P99.5 for MRQNBC", but I found in figure 3 that the corresponding value for MRQNBC is quite small. Therefore, I'm quite confused by many expressions in the current manuscript.

**Response:** This was an incorrect statement and we thank the referee for pointing this out. After revising the manuscript, we will make sure to properly proofread the manuscript, so there won't be any incorrect and confusing expressions left.

---

## Author Response (AR1)

**Response to Referee Comment 1 on 'Impact of bias nonstationarity on the performance of uni- and multivariate bias-adjusting methods' by J. Van de Velde et al.**

**Bastien François (Referee)**

We would like to thank Bastien François for his fair and thorough review. Below, we give a comment-by-comment response.

**General Comments**

Comment: Results are often not well explained when bad performances for (M)BCs are obtained. The validation metrics used in this study (residual biases relative to the observations (Rbo) and to the model bias (Rbmb)) tend to make, in my opinion, the analyses of results difficult for readers. In that respect, I would find helpful to provide raw values of the indices for, respectively, the calibration and projection periods (i.e. without computing directly their biases as in Table A1) and their changes between the calibration and projection periods for: the reference, the simulations and the 5 corrections from the (M)BC methods. In a general way, it would permit to better explain the results obtained from the (M)BC methods, i.e. whether good or bad results for a specific method come from the nonstationary problem (i.e. from the fact that simulated changes are "wrong", which alter the quality of results from MBC methods) or from the characteristics of the method (e.g., the statistical technique used, its stochastic nature, etc.). Identifying the reasons for good or bad results from (M)BC methods is of key importance for this study, to be able to conclude with more certainty that bias nonstationarity is the main source of problem.

**Response:** We used the RBo and RBmb metrics as a method to present detailed information on the full extent of changes in a visual way and to enable the reader an easier comparison without needing too many tables. However, these are new metrics and we agree that the raw values are a valuable source of additional information that can provide additional guidance to the reader. However, as referee #2 preferred seasonal or monthly data, providing all raw values and biases would have seriously lengthened the paper. As the RBo and RBmb values can help understand the bias adjustment behavior at some points, we preferred to keep those in the paper and to provide additional information where necessary.

Comment: The authors conclude that "the univariate bias-adjusting methods, computationally less complex and not taking (potentially changing) correlations into account, seem to be more robust." (L787). Concerning correlation, I am afraid that this result of robust-ness is only specific to the present application, and hence the generalization of this conclusion cannot be done as the authors do (e.g., L825 "we advise to use univariate bias-adjusting methods, until it becomes more clear how it can be ensured that multivariate methods certainly perform well in a climate change context."). Indeed, one of the advantages of considering MBC methods instead of univariate ones is that MBCs are able to adjust correlations between variables. Univariate BC methods are not designed to do so. For example, 1D-BC methods such as QDM globally conserve the rank structures of the climate model to correct, and hence the simulated dependence structure between variables is preserved. According to Table A1, simulated correlations often present little bias compared to observations (for example, for corr(P,E), corr(E,T), crosscorr(P,E,0), crosscorr(E,T,0), crosscorr(E,T,1)). Then, by preserving the simulated rank structures, 1d-BC methods like QDM mechanically present correlations with little bias as well. This result is really specific to this study, and would not be obtained if the raw climate simulations would have presented strong biases in correlations. Hence, concerning correlations, results from 1d-BC as QDM depend on how well the models simulate relevant dependencies between the climate variables. This is something already suggested by Zscheischler et al. (2019). This point is not explained in the present study, while it is one of the key points of discussion, and the principal reason why QDM often performs well in this study for

correlations. This point should be mentioned and discussed to provide the appropriate nuances to initial conclusions.

**Response:** The bias in correlation and the simulation of dependencies is a point that should have been mentioned better. We addressed this in the revised manuscript in the section on correlation results and the Discussion and conclusions section.

Comment: Linked with the comment 1. above, I would find interesting to provide (at least in Appendix) the results for the multivariate bias correction and the hydrological model for the calibration period. In my opinion, verifying that MBC methods perform well for the calibration period (which has to be verified) would validate the global methodology and would better support the conclusions on the effect of bias nonstationarity on the results from MBCs for future periods.

**Response:** We have calculated the adjustment for all methods in the calibration period. The information will be included in the revised version. This shows, in combination with the seasonal results requested by referee #2 that the original statement that multivariate bias-adjusting methods perform poorer than the univariate bias-adjusting methods was an overstatement. Nonetheless, the comparison illustrates that 1) for some seasons, there is a clear influence of bias nonstationarity on the results and 2) that the multivariate bias-adjusting methods are less well equipped to deal with the seasonal differences in bias nonstationarity.

Comment: As explained by the authors (L79), climate models can present statistical biases compared to observations that are nonstationary, that is, that the differences of bias between the calibration and future period are not the same. What is often not clarified clearly through the article is that, in a changing climate context, another way to see bias nonstationarity is that it results from the fact that observed and simulated variables do not present the same changes between the calibration and projection periods. This is of key importance, as some of the (M)BC methods included in this study can take into account the simulated changes in their correction procedures (such as dOTC or MBCn). It must be better highlighted through the article, and has to be used to provide a better analysis of the results.

**Response:** This information was added to the introduction of Section 3.3 and considered in the Results and Discussions and conclusions sections where necessary.

Comment: I think the reasons why the three MBCs (MRQNBC, dOTC and MBCn) are selected in this study must be better specified. In particular, their differences of assumptions and attributes could be better indicated, for example with a table (as recommended in 6.).

**Response:** The introduction for the multivariate bias-adjusting methods was extended and an overview table was added to the end of this discussion. With this table, the reader should be able to get a clearer overview of the differences between the methods.

Comment: If I understood well, all the three MBC methods are able to take into account (some of) the potential simulated changes in their correction procedures. It also exists in the literature MBC methods that assume dependence structure to be stable in time (R2D2, Vrac, 2018, Vrac and Thao, 2020). It would have been interesting to include in the study such MBC methods as benchmarks for a better assessment of the potential losses (or benefits) of considering stability of dependence structure, even in a changing climate context.

**Response:** The R2D2 method is indeed a valuable method to include in the analysis. We have now included the method. In addition, Section 3 has been extended to give a good overview of this method.

Comment: The quality of presentation should be improved.
- The "Bias-adjusting methods" section is too long. Please consider to summarize each of the BC methods and their main properties in a short text (and with the help of a table, as already explained)

to keep from overwhelming the reader with technical details that are not necessarily useful to get the main points. Technical details and algorithms can eventually be placed in Appendix.
- Please consider to indicate the letters for each figure when describing the results (for instance, for QDM: Figure 3a, mQDM: Figure 3b, etc). This would better guide the reader through the different plots.

**Response:** In the revised paper, the technical details have been cut from the paper. Adding an Appendix would've increased the length of the paper too much. As mentioned previously, a table was added in the subsection on multivariate methods to better guide the reader. Where needed, the figure panel was specified to better guide the readers.

**Specific Comments**

Comment: L55: I would replace the word "uncertainty" as it can be misleading here. I would rather say "error (or bias) that can propagate in the impact models".

**Response:** Thank you for the suggestions, this was updated. (L 52-53 in the revised paper).

*If the correlation between these variables is biased w.r.t. the observations, then it can be expected that the model output is biased as well, which can further propagate in the impact models.*

Comment: L72: in François et al. (2020), it is explained that all multivariate methods that are under study fail in adjusting the temporal structure of simulated times series. Of course, it already exists methods that can adjust (part of) the temporal properties: MRQNBC is one of them.

**Response:** The sentence was improved to better reflect this nuance.(L 70 in the revised paper)

*Besides, they also noticed that all multivariate methods studied fail in adjusting the temporal structure of a time series.*

Comment: L75: what do you mean by "more recent validation period". I think that just mentioning "validation period" is enough.

**Response:** This was updated in the paper. (L 73 in the revised paper)

Comment: L225: Did you verify that assuming the stationarity of the frequency of dry days holds for your application? Also, did you apply a thresholding after bias correction? I know that, for example, dOTC can generate negative values for precipitation. How did you deal with this problem?

**Response:** As is shown in the results (R index values) the number of dry days can be considered stationary for the time periods studied. A reference to the first Section of the results was added (L 224 in the revised paper). For dOTC, we applied a thresholding before calculating the indices, as this was suggested in Robin et al. (2019). However, because of space limitations, the technical details are not included in the paper and hence this will also not be mentioned.

*However, as thresholding is used prior to all methods, the influence of possible bias nonstationarity on $\Delta N$ is assumed to be negligible. Besides, as is shown in Section 4.1 the number of dry days is stationary for the time frames studied in this paper.*

Comment: L282: The successive conditional approach performs successive corrections conditionally on the variables already corrected. This can be applied to more than two variables. It should be specified. It does not "adjust a second variable conditionally on the second variable", as written.

**Response:** This sentence was indeed incorrect. We have clarified that it can be applied to more than two variables. (L 276 in the revised paper)

*These two components are then recombined to obtain data that are close to the observations for both marginal and multivariate aspects. The latter approach consists of adjusting a variable conditionally on the variables already adjusted. This procedure is applied successively to each variable.*

Comment: L286: This is not well explained. The problem of robustness of the successive conditional approach is specific for a high number of dimensions to correct. Indeed, in a successive conditional approach, as the number of variables already corrected increases at each successive step, it progressively reduces the number of data available for the correction, making it less and less robust.

**Response:** As successive conditional methods are not studied in this paper, these sentences were removed in the revised paper.

Comment: L396: Do you know why the results are exacerbated?

**Response:** The first repetition created some unphysical values (especially for precipitation). As the implementation is based on the correlations, every repetition exacerbated these nonphysical values. However, when running the method just once, the results were satisfactory. This was updated in the paper (L 335-340 in the revised paper). The method is not entirely suited for bounded variables such as precipitation and evaporation. Nonetheless, as some of the results were good in both calibration and validation periods, methods based on the same principles are worth investigating and using for precipitation.

*However, the nesting method cannot fully remove biases at all time scales, thus Mehrotra et al. (2016) suggested to repeat the complete procedure multiple times. Yet, in our case multiple repetitions exacerbated the results. Nonphysical outliers created by the first repetition influenced the subsequent repetitions, creating even more nonphysical values. This was most clearly seen for precipitation. As a bounded variable, precipitation is most sensitive for nonphysical values. Nonetheless, running the method just once yielded agreeable results.*

Comment: L429-440: I am not sure that the equations to explain the Schaake Shuffle are necessary, as it can simply be explained with words.

**Response:** Not every written explanation of the Schaake Shuffle is equally clear, and we wanted to ensure that readers could follow by providing an exact mathematical formulation. However, these technical details have been removed from the paper to limit the length.

Comment: L442: Do you have a reference to account for ties by introducing small random values? There exists also other ways to compute ranks to handle problems of equal values.

**Response:** This was based on the approach used in Vandenberghe et al. (2010), in which ties were removed before copulas were fit. Vandenberghe et al. (2010) based this on the discussions in Salvadori and De Michele (2006) and Salvadori et al. (2007). However, these details were removed from the paper, as mentioned earlier.

Comment: L445: This result is really "case-specific". I would precise it, as it cannot be generalized for each application (depending on the bias-nonstationarity, for example).

**Response**: These sentences were removed from the paper to limit the length.

Comment: L454: Did you check if, even after early stopping, overfitting was not a problem? In particular, how did you choose the tolerance of 0.0001?

**Response**: We checked overfitting by studying the final distance scores: these show that there is still some margin for improvement. The tolerance was chosen by trial and error: this level of tolerance yielded fairly good results while not increasing the duration too much. This was not added to the revised paper, as the technical details were removed.

Comment: L463: I would precise that dOTC extends the CDFt method to the multivariate case.

**Response:** This was added to the paper.(L 370-371 in the revised paper)

Comment: L465: dOTC is indeed designed to preserve the trend of the model for marginal properties but also for dependence structure (or copula). It should be specified, as it is a major difference with univariate methods.

**Response:** This specification is indeed important and was added to the paper. (L 373 in the revised paper)

*The combination of both optimal transport plans allows for bias adjustment while preserving the trend of the model* *for both marginal properties and the dependence structure.*

Comment: L520: I don't think that the 10 calculations made for dOTC are necessary. What are the bin sizes chosen to implement dOTC? If the bin sizes are small, the influence of the stochastic components in dOTC is rather small, and hence introducing 10 calculations does not seem necessary to me.

**Response:** For each variable, 25 bins were used. More bins quickly increased the computational load. The bin size depends on the variable space. The results differed among the calculations, and thus it is better to keep them and average the results.

Comment: L593: I do not understand the sentence "Yet, QDM has the best RBMB values and might thus be preferable", that seems in opposition with the previous sentence.

**Response:** The results were rewritten based on seasonal results (a request of Referee #2) and this sentence was removed.

Comment: L644: "an influence" on what? Correlations? Please be more precise.

**Response:** The results were rewritten based on seasonal results (a request of Referee #2) and this sentence was removed.

Comment: L677: Actually, it surprises me as it has been found in François et al. (2020) that, for a small number of dimensions to correct, MBCn and dOTC preserve roughly the rank sequences of the model to correct. It normally does not have such "important shuffling" as specified. Do you know why?

**Response:** This should have been written differently. As can be seen in Figure 7, the RBmb and RBo values for the lag-1 autocorrelation and wet-dry transition probability (panel (c), MBCn) and the number of dry days (panel (e), dOTC) are close to 1, indicating that they are very close to the original climate simulation values. Thus, the methods indeed roughly preserve the rank sequence. As the results were rewritten based on the seasonal results, this sentence was removed. However, ensured that the writing is clearer on this specific aspect of the method.

Comment: L698: A difference between QDM and MBCn is also the adjustment of dependence structure. It should be specified.

**Response:** Thank you for addressing this point. Although the specific sentence was removed, this point was considered when revising the Results section.

Comment: L705-706: I do not understand. Please rewrite.

**Response:** The results were rewritten based on seasonal results (a request of Referee #2) and this sentence was removed.

Comment: Did you mean "possibly"?

**Response:** Yes, thank you for noting this mistake.

**References**

Salvadori, G. and De Michele, C.: Statistical characterization of temporal structure of storms, Advances in Water Resources, 29, 827–842, https://doi.org/10.1016/j.advwatres.2005.07.013, 2006

Salvadori, G., De Michele, C., Kottegoda, N. T., and Rosso, R.: Extremes in nature: an approach using copulas, vol. 56, Springer Science &Business Media, 2007

Vandenberghe, S., Verhoest, N. E. C., and De Baets, B.: Fitting bivariate copulas to the dependence structure between storm characteristics: A detailed analysis based on 105 year 10 min rainfall, Water resources research, 46, https://doi.org/10.1029/2009WR007857, 2010

**Response to Referee Comment 2 on 'Impact of bias nonstationarity on the performance of uni- and multivariate bias-adjusting methods' by J. Van de Velde et al.**

**Anonymous referee**

We would like to thank the referee for the time spent reviewing our manuscript. Below, we give a comment-by-comment response.

**General Comments**

Comment: The authors did not explicitly present the study area of this paper. After looking for a while, I'm surprised to find that only one station and one grid cell were used (lines 115-143). In addition, there is only one GCM-RCM combination was used, and no information on its spatial resolution. As such, the results of this study are subjected to large uncertainty.

**Response:** We believe our research design is still informative, as it is too date scarcely addressed and can provide a base-line for more extended follow-up studies. Yet we simultaneously agree on the limitations of our design, and we have now properly stated this in both the Introduction and the Discussion.

Comment: The manuscript is lengthy and hard to follow. I think some information does not need to be presented in detailed in the manuscript. For example, the introduction of the bias correction methods (section 3.1 to 3.3, almost 14 pages of the main body). All these information can be found in literatures.

**Response:** We have removed the technical information and other minor unnecessary parts of the paper, such as the information on climate change in Uccle. Besides, we integrated both 'Discussions' and 'Conclusions' sections in one 'Discussion and conclusions' section

Comment: Although the authors have stated in line 527 "As the effect on discharge is the overarching goal of this paper", I think the information on the hydrological model and hydrological simulation this quite poor in the manuscript. Firstly, I did not find how the hydrological model performed in the study area (e.g. Nash value or some other criteria). Although the goal of this study is to compare the difference between the univariate method and multivariate method in the hydrology perspective, the hydrological model should be well calibrated. Secondly, as is stated that the PDM model was not calibrated in Uccle but Grote Nete watershed, please give the evidence to show that it is feasible to drive the PDM using the climate data in Uccle.

**Response:** Some details on the calibration performance were added. Besides, the Grote Nete watershed is roughly 50 km from the Uccle station and the effect of topography on weather is negligible. Hence, the assumption that the Uccle data can be used for the watershed is acceptable, as discussed in the referenced and other earlier papers. (L. 185-196 in the revised paper)

Comment: The authors used scatter plots throughout the whole paper. It is difficult to see how many dots are located within the 0-1 square or to do the comparison (e.g. Figure 6). I suggest that maybe the authors can use some more quantitative metrics and figures to show the results.

**Response:** The scatter plots allowed for the inclusion of multiple indices or interesting percentiles of the variables' distribution in one plot and are hence practical for conveying information. However, we understand that using only the RBmb and RBo values might impede some readers in quickly grasping the results. As such, we have include other measures, such as Perkins Skill Score and ETCCDI indices, wherever applicable. The inclusion of these measures allowed to give a more balanced overview of the results.

Comment: Figure 3 to figure 9, the authors did not mention which period (calibration or validation) and which month the results are based, or is it based on the whole year? In this case, readers cannot understand and assess the results. For example, for the correlations between two variables (e.g. P and T), the correlation coefficients are quite different for each month (e.g. summer and winter), therefore, at least, they should be evaluated separately for each month.

**Response:** These figures were based the validation period, with data for the whole year. All figures are updated to clarify the time frame. Besides, as suggested, we have calculated monthly and seasonal data. This indeed allows for a better assessment of the inter-seasonal differences, which provides important insights in the propagation of bias nonstationarity. To provide a balanced overview of the results and keep a proper length of the paper, we preferred the seasonal evaluation. Hence, all results are rewritten to reflect the updated validation.

Comment: Table 1, I am wondering whether the Spearman correlation coefficients and lag-0 cross-correlation between two variables reflect the same thing.

**Response:** The Spearman correlation is based on the rank of the value, the crosscorrelation is based on its effective value; these two values thus do not reflect the same thing. This can also be seen in Figure 8 of the revised paper, where the RBmb and RBo values for the Spearman correlation and the always differ by some extent.

Comment: In general, I found that many expressions in the current manuscript are not very accurate. For example, in line 568 the authors write "A surprising result for P is the high RBMB value for P99.5 for MRQNBC", but I found in figure 3 that the corresponding value for MRQNBC is quite small. Therefore, I'm quite confused by many expressions in the current manuscript.

**Response:** This was an incorrect statement and we thank the referee for pointing this out. After revising the manuscript, we made sure to properly proofread the manuscript.

---

## Referee Report (RR1)

This manuscript presents an attempt by the authors to investigate the effect of non-stationary biases may have on the performance of multivariate bias-adjusting methods for regional climate models. To do so they have used four multivariate methods (MBCa, MRQNBC, $R^2D^2$, dOTC) and two univariate ones (QDM, mQDM), to adjust bias in the output from a single GCM-RCM run (12.5km RCA4 forced with boundary conditions from the MPI-ESM-LR global model) for three climate variables (temperature, precipitation, and evaporation). Precipitation and evaporation have been used to drive a rainfall-runoff model as a test of the impact that bias adjustment can have on variables used for impact studies. The authors conclude that non-stationary biases are important for bias adjustment procedures without reaching a firm conclusion on the issue of the relative performance of uni-and multivariate bias adjustment. I believe this is the result of poor methodological choices by the authors, which constrained their ability to reach a more meaningful conclusion in what is an interesting and relevant topic. More specific comments regarding the methods follow:

1. All the bias adjustment calibrations and validations were carried out using model output from a single model cell, of a single RCM-GCM combination, with a single observed dataset as reference. Therefore, the results presented in the paper may be unrepresentative of the behaviour of the bias adjusting methods, which could be explored much more robustly by exploring multiple locations and models. The data choices give the authors a single comparison point for the bias correction methods, a larger sample would help reduce the uncertainty present in the results and possibly lead to stronger conclusions.

2. The results are unclear; they lack clear trends and demonstrate the limitation of the single-location approach as each seasonal index is plotted once and therefore no conclusion can be drawn as to how representative the results really are. While the indices used are useful in representing different portions of the distribution of each variable, no statistical evaluation of the significance of the differences was attempted, either through the use of summary statistics or graphically.

3. The emphasis given to hydrological models in the abstract is lost throughout the paper. Very little detail is provided about the rainfall-runoff model in the methods, in particular there is a single RMSE value as sole evidence of model calibration with the reader referred to a previous paper for details despite this being a key aspect of the paper. In addition, the data point used to evaluate the bias methods lies outside the model catchment and no evidence is provided to support the similarities between the sites.

---

## Referee Report (RR2)

In the third draft, the authors of "Impact of bias nonstationarity on the performance of uni- and multivariate bias-adjusting methods: a case study on data from Uccle, Belgium" took into account all my comments in a satisfactory way, including 1) the improvement of the design experiment to compare the different bias correction methods properly, 2) the description of Rank Resampling for Distributions and Dependences, 3) the improvement of section 4.1 "Bias change". These modifications improve the manuscript. I appreciate the work done by the authors to modify the initial draft.

However, the second reviewer mentioned the issue that the evaluation has been done only for one cell and a single model. I agree with the second reviewer that this issue is key to (potentially) reach a firm conclusion on the performance of univariate and multivariate bias correction methods, which is not the case in the present study. Indeed, drawing conclusions on the performance of these statistical methods using only one grid point seems to be subject to much uncertainty.

From my point of view, this point is so important and should be considered in a future version, to provide clearer conclusions to end-users.

**Specific comments:**

- Figure 1: the panel (f) is missing.
- After the different modifications during the peer-reviewing process, I found the final submission very detailed, which is quite interesting, but makes it sometimes difficult to read. From my personal point of view, an effort to be concise must be made, so that readers do not have problems following the manuscript. In particular, are there strong differences of conclusions between 4.2-4.4? Would gathering the results and summarizing concisely the conclusions obtained possible (and preferable)?
- I often found it hard to link the information given by the subsection 4.1 Bias Change, and the other sections 4.2-4.7 giving the results for the performance of the methods while this link is the core of the study.

---

## Referee Report (RR3)

Reviewer 2 report - hess-2020-639

General comments:

The reviewer appreciates the work made by the authors to improve the manuscript, this includes additional framing of the results and the work as a case study of value, with much improved justification of the choice and discussion of the limitations. The additional results tables provide much needed clarity to the results. My only general comment would be to urge the authors to limit their use of long paragraphs where important ideas can get lost to the reader (two examples of this are included in the specific comments). Some minor comments follow.

Specific comments:

L 19-20: To which adjustment do you refer here?

Table 3: $P_5$ is NAN in all seasons, why? Should it remain in the table?

L 405: Consider separating table 3 and table 4 results into two paragraphs.

Figure 1: Panel f is missing.

L 437: Why can't these percentiles be plotted, is it by choice? If this is the case, a reference in the text justifying the decision to limit the figure's range would be useful.

Figure 6 caption: $R^2D^2$ tag is missing next to (f).

L 528: It could be worth discussing that there is a seasonal component to the effect of ETP, could winter biases be more relevant for flood modelling than summer biases in ETP?

Table 8: mQDM, Winter, Cal: value is 1%, should this be 100%?

L 627-660: This is a very long paragraph which can be split to ease reading (e.g., along sentences starting with First, Second, Third…).

L 648: Related to the above, the sentence starting here is important and should be highlighted as an important result (perhaps by starting a new paragraph?)

---

## Author Response (AR2)

**Response to Referee #1 (Bastien François) on 'Impact of bias nonstationarity on the performance of uni- and multivariate bias-adjusting methods' by J. Van de Velde et al.**

In the second draft, the authors of "Impact of bias nonstationarity on the performance of uni- and multivariate bias-adjusting methods'' took into account part of my comments, as well as comments from the second reviewer. In particular, it results in evaluating the performance of 2 univariate and 4 multivariate BC methods under climate change conditions in order to determine the influence of bias nonstationarity on results. Instead of evaluating their results over the whole year (as in the first draft), the authors followed the advice from the second reviewer and performed a seasonal evaluation. They conclude that non-stationarity can have an important influence on the performance of the MBC methods and the propagation of biases in impact models. The authors found that the importance of the influence varies depending on seasons and variables. They finally advise to account for seasonality for a robust bias adjustment under bias-nonstationarity, and advice to use univariate methods instead of multivariate ones until it becomes more clear how MBCs perform under bias nonstationarity.

While I appreciate the work done by the authors to modify the initial draft and take into account the comments from the reviewers, I think that several major limitations still remain in the present study and can be improved.

**General comments:**

1) The design experiment does not permit to assess properly the influence of non-stationarity on the performance of the MBC methods and should be modified. If I understood well, the 2 univariate BC (QDM and mQDM) are applied over a 91-day moving window, while the 3 multivariate BC (MBCn, R2D2 and dOTC) are applied over the full time period. Applying the 3 multivariate BC (MBCn, R2D2 and dOTC) over the full time period and evaluating them at a seasonal scale is not appropriate, and hence presents a major issue for the interpretation of the results. Indeed, as pointed out by the authors in Section 4.1, biases can vary considerably depending on the season: for example, a climate model can present very little bias in winter and be drastically biased in summer. Thus, the correction of the statistical properties of the model (such as mean, variance, correlations) can be very different depending on the season. By applying MBC methods such as MBCn, R2D2 and dOTC that do not include seasonal components in their procedure over the full time period, it generates data that potentially present bias introduced by the design experiment: the MBC methods would correct the different seasons by applying a similar statistical transformation. Consequently, model data for seasons with strong biases won't be corrected enough, whereas model data for seasons with little biases could be deteriorated. In the study, applying these 3 methods (MBCn, R2D2 and dOTC) as such potentially introduces a bias by construction, placing them at a disadvantage in the intercomparison study compared to QDM and mQDM. Moreover, it makes it impossible to identify if bad performances from these MBCs are due to either bias nonstationarity or artefacts from the design experiment. This problem is known by the authors, as discussed in L638 (« It is thus unclear whether the poor seasonal performance obfuscates the effect of nonstationarity, or if the similar performance is a sign of robustness. ») and L640 (« Hence, the set-up does not allow to clearly discern between the various categories of multivariate bias-adjustment, such as the 'marginal/dependence' or 'all-in-one' categories. »). But assessing the effect of nonstationarity on the performance of MBCs is initially the main objective of the study. The design experiment must be established in order to isolate the effect of nonstationarity as much as possible. For example, it can be done by applying all the BC methods that do not include seasonal components in their correction procedure (QDM, mQDM,

dOTC, R2D2 and MBCn) over the same seasonal period (e.g. over winter or summer, but separately), and then performing seasonal evaluations to fairly compare them. It would permit the intercomparison of BC methods, all things being equal.

**Response:** We have updated the methods to work with seasonal input. This has, as suggested, an impact on the results. Consequently, the Results and Discussions and Conclusions sections of the text were rewritten to reflect these changed results.

2) The new implementation of a MBC method, named Rank Resampling for Distributions and Dependences (R2D2, Vrac et Thao, 2020), is unclear. This MBC method relies on an analogue-based technique for which some conditioning information is required to adjust dependence structure of the simulated time series. The conditioning information can be multivariate, by considering a set of variables at a given time t. It can also be extended to ranks sequences, i.e. conditioning by not only one but several lagged time steps. The choice of the conditioning information is crucial to interpret the results from R2D2, as it can have impacts on marginal, inter-variable and/or temporal properties. This information, and its influence on bias-corrected data from R2D2, is not precisely given in the paper. Consequently, results from this MBC method cannot be analyzed in an appropriate way by the readers. Moreover, at L362 is indicated: « Each variable (precipitation, evaporation and temperature) was in turn used as the reference dimension. ». This implies that 3 bias-corrected data were produced for R2D2, but, surprisingly, only one result for R2D2 is presented in the study. Thus, further clarification is required to better present the results from the R2D2 method.

**Response:** The information on R2D2 has been extended to be clearer and to better reflect our application.

*In the present application of R2D2, QDM was used as the univariate bias-adjusting method to ensure consistency with the other multivariate bias-adjusting methods. This ensures the preservation of the changes in the marginal distribution. Each variable (precipitation, evaporation and temperature) was in turn used as the reference dimension. As the present study was limited to a single grid cell, the use of additional data was limited. However, to ensure that the selection of analogues is diverse enough, five lags were used to search for analogues, three of which were retained in the resampling. Finally, the results for the three variables were averaged to present the final R2D2 result.*

3) The Section 4.1 'Bias change' is hard to read. Is a table missing? The authors describe index values for bias change between the calibration and projection period, but a table seems necessary to better present the results and facilitate reading.

**Response:** A table has been added.

4) I would like to thank the authors for providing the results for the calibration period. However, linked with my first comment, it highlights that MBCn, dOTC and R2D2 methods are not applied in an appropriate manner compared to QDM and mQDM: For example in Table 4, PSS values indicate poor performances for these 3 MBC methods during the calibration period, principally because they are applied over the full time period and evaluated by seasons. If MBC methods do not produce good results on the calibration period on indices that are supposed to be adjusted, then MBC methods are not well "calibrated" and no good results can be expected for the validation period. This point should be considered if my first comment is taken into account.

**Response:** Applying the multivariate methods on a seasonal basis has clearly improved the PSS results, and this is now reflected in the text. In case the PSS value is not similar among the methods, this is taken into account in the discussion of the results.

5) Also, linked with 4), the advantages of the "marginal/dependence" methods such as MBCn is that, for evaluation criteria on marginal properties such as PSS in Table 4, same performances must be obtained between QDM and MBCn (trend preservation), by construction. It would be nice to consider retrieving these results on marginal properties before analysing other indices, such as correlation or discharge.

**Response:** As described for 4), the performance of the multivariate methods, and hence MBCn, improved with seasonal inputs. The PSS values for MBCn are now (for the marginal properties) the same as those for QDM, indicating that these methods can be compared for other indices.

**Specific comments:**

6) L318: In this study, dOTC is not the most recent method used, but R2D2 2.0

**Response:** This has been adjusted. We now refer in this sentence to dOTC as 'the last method' (as mentioned in the paper).

7) L526 « Both the univariate and the multivariate bias-adjusting methods can adjust the simulated biases well » and L530 « the good adjustment by univariate methods is trivial: they will adopt the correlation of the simulations and only slightly adjust this by adjusting the marginals. » I was wondering if you can rephrase these sentences in order to avoid saying that univariate bias-adjusting methods adjust correlations. Improvements of correlations are only due to an indirect effect of the adjustments of marginal properties.

**Response:** The first sentence has been removed, while the second was slightly rephrased.

8) L527: « The univariate methods will adopt the dependence structure of the raw simulations » I am not sure if it is true for mQDM, that will have exactly the same rank correlation structure than the observations by construction (at least for the calibration period).

**Response**: This has been adjusted to refer to only QDM.

9) Table 2: As requested, a table is introduced in order to summarize the different characteristics of the MBC methods. This table can be very useful for the readers, but the actual one presents some formulations that are not clear enough or misleading. Some examples: for the row « Temporal properties » and column « dOTC », the information « Future, adjusted » is misleading. dOTC is not designed to adjust temporal properties and must be clearly indicated. Another formulation must be used instead to add more nuances and to specify that potential unexpected behaviors of temporal properties can be obtained with dOTC. For the column « R2D2 », the information « Shuffle based on observations » is not clear enough: temporal properties of the bias-corrected data depend on the conditioning information used (see my second point). For the column «MBCn», the information « Shuffle based on observations » is wrong: temporal properties from the model are modified in an uncontrolled manner by the decorrelation/recorrelation procedure and the univariate correction. However, empirical findings in François et al., 2020 indicate that MBCn (and hence the decorrelation/recorrelation procedure) tends to conserve partially the rank sequences from the model, in particular in the context of bias-correction of a small number of statistical dimensions. Moreover, it might be necessary to change the order of the rows in order of importance. I find it odd

to have the row « Temporal properties » at the beginning of the table and « Statistical technique » almost at the end of the table.

**Response:** The table has been adjusted based on your comments. The information was reordered and some of the characteristics were updated to better reflect the specifications of the methods. In addition, some of these points are now better clarified in the main text.

**Response to Referee #2 (Anonymous referee) on 'Impact of bias nonstationarity on the performance of uni- and multivariate bias-adjusting methods' by J. Van de Velde et al.**

This manuscript presents an attempt by the authors to investigate the effect of non-stationary biases may have on the performance of multivariate bias-adjusting methods for regional climate models. To do so they have used four multivariate methods (MBCa, MRQNBC, $R_2D_2$, dOTC) and two univariate ones (QDM, mQDM), to adjust bias in the output from a single GCM-RCM run (12.5km RCA4 forced with boundary conditions from the MPI-ESM-LR global model) for three climate variables (temperature, precipitation, and evaporation). Precipitation and evaporation have been used to drive a rainfall-runoff model as a test of the impact that bias adjustment can have on variables used for impact studies. The authors conclude that non-stationary biases are important for bias adjustment procedures without reaching a firm conclusion on the issue of the relative performance of uni-and multivariate bias adjustment. I believe this is the result of poor methodological choices by the authors, which constrained their ability to reach a more meaningful conclusion in what is an interesting and relevant topic. More specific comments regarding the methods follow:

1) All the bias adjustment calibrations and validations were carried out using model output from a single model cell, of a single RCM-GCM combination, with a single observed dataset as reference. Therefore, the results presented in the paper may be unrepresentative of the behaviour of the bias adjusting methods, which could be explored much more robustly by exploring multiple locations and models. The data choices give the authors a single comparison point for the bias correction methods, a larger sample would help reduce the uncertainty present in the results and possibly lead to stronger conclusions.

**Response:** Although the data is indeed limited, we believe that it is still possible to derive useful results. Considering the comments of Referee #1, the results clearly illustrate how biases can be influenced by bias nonstationarity, and how this can propagate to an impact model. Yet, we acknowledge that this is only a case study and would like to present our study as such. Nevertheless, this provides a framework that other studies could expand upon. In addition, we have expanded the discussion to compare the climate models used here with other models from the EURO-CORDEX ensemble. This provides only a limited overview, but still allows other researchers to place this study within a broader context.

2) The results are unclear; they lack clear trends and demonstrate the limitation of the single-location approach as each seasonal index is plotted once and therefore no conclusion can be drawn as to how representative the results really are. While the indices used are useful in representing different portions of the distribution of each variable, no statistical evaluation of the significance of the differences was attempted, either through the use of summary statistics or graphically.

**Response:** It was our goal to indicate the significance using the RB_MB and RB_O metrics. Whereas RB_MB indicates the effectiveness of the bias adjustment, RB_O is an indicator of the significance in comparison with the observations. However, the explanation of these metrics was not clear enough and has now been extended. In addition, we have paid more attention to this throughout the text. Also, the comments of

referee #1 helped to clarify the results. Finally, the goal of this manuscript is not to clarify trends, but to give an overview of how bias-adjusting methods might respond to bias nonstationarity.

3) The emphasis given to hydrological models in the abstract is lost throughout the paper. Very little detail is provided about the rainfall-runoff model in the methods, in particular there is a single RMSE value as sole evidence of model calibration with the reader referred to a previous paper for details despite this being a key aspect of the paper. In addition, the data point used to evaluate the bias methods lies outside the model catchment and no evidence is provided to support the similarities between the sites.

**Response:** As we wanted to study the sensitivity of an impact model to biases and bias nonstationarity, small discussions of the (possible) impact of bias nonstationarity on the marginal properties, correlation and occurrence are now included in each Results subsection. This allows for a better understanding of the propagation of the bias nonstationarity and the role of the hydrological model. However, as indicated in the abstract, the hydrological model serves only as an illustration and the main goal is understanding the sensitivities of an impact model to bias nonstationarity. As such, adding more information on the calibration does not seem necessary, as it is not a key aspect for us. The issue of the distance between the climate model grid cell and the catchment was already discussed in the text, albeit limited.

---

## Author Response (AR3)

**Response to Referee Comment 1 on 'Impact of bias nonstationarity on the performance of uni- and multivariate bias-adjusting methods' by J. Van de Velde et al.**

**Bastien François (Referee)**

We would like to thank Bastien François for his final comments on this paper. Below, we give a comment-by-comment response.

**General Comments**

Comment: In the third draft, the authors of "Impact of bias nonstationarity on the performance of uni- and multivariate bias-adjusting methods: a case study on data from Uccle, Belgium" took into account all my comments in a satisfactory way, including 1) the improvement of the design experiment to compare the different bias correction methods properly, 2) the description of Rank Resampling for Distributions and Dependences, 3) the improvement of section 4.1 "Bias change". These modifications improve the manuscript. I appreciate the work done by the authors to modify the initial draft.

However, the second reviewer mentioned the issue that the evaluation has been done only for one cell and a single model. I agree with the second reviewer that this issue is key to (potentially) reach a firm conclusion on the performance of univariate and multivariate bias correction methods, which is not the case in the present study. Indeed, drawing conclusions on the performance of these statistical methods using only one grid point seems to be subject to much uncertainty.
From my point of view, this point is so important and should be considered in a future version, to provide clearer conclusions to end-users.

**Response:** We thank the reviewer for his appreciation of our work. In addition to the better framing of the choices that was added in the previous review round, some ideas on how to assess the bias nonstationarity on a continental scale have been added.

**Specific Comments**

Comment: Figure 1: the panel (f) is missing.

**Response:** We thank the reviewer for noticing this. The figure has been updated.

Comment: After the different modifications during the peer-reviewing process, I found the final submission very detailed, which is quite interesting, but makes it sometimes difficult to read. From my personal point of view, an effort to be concise must be made, so that readers do not have problems following the manuscript. In particular, are there strong differences of conclusions between 4.2-4.4? Would gathering the results and summarizing concisely the conclusions obtained possible (and preferable)?

**Response:** We thank the reviewer for indicating lack of clarity. To improve this, the last paragraph of each subsection has been expanded to summarize the results for the corresponding variable or measures. In addition, the second paragraph of the Discussions and conclusions section has been extended to give a global summary of the results.

Comment: I often found it hard to link the information given by the subsection 4.1 Bias Change, and the other sections 4.2-4.7 giving the results for the performance of the methods while this link is the core of the study.

**Response:** This information was given throughout the text, but to further improve clarity, the extended paragraphs mentioned in the response to the previous comment also allowed for a better discussion of the link between the R index values and the performance of the methods.

**Response to Referee Comment 2 on 'Impact of bias nonstationarity on the performance of uni- and multivariate bias-adjusting methods' by J. Van de Velde et al.**

**Anonymous referee**

We would like to thank the referee for the additional comments on this paper. Below, we give a comment-by-comment response.

**General Comments**

Comment: The reviewer appreciates the work made by the authors to improve the manuscript, this includes additional framing of the results and the work as a case study of value, with much improved justification of the choice and discussion of the limitations. The additional results tables provide much needed clarity to the results. My only general comment would be to urge the authors to limit their use of long paragraphs where important ideas can get lost to the reader (two examples of this are included in the specific comments). Some minor comments follow.

**Response:** We thank the reviewer for the appreciation of our work. In addition to the paragraphs mentioned in the specific comments, the comments by Referee #1 have also led to more clarity and some improved paragraphs.

**Specific Comments**

Comment: L 19-20: To which adjustment do you refer here?

**Response:** This refers to the adjustment (by all methods) in the validation period, in comparison with the adjustment in the calibration method. The sentence has been updated.

Comment: Table 3: P5 is NAN in all seasons, why? Should it remain in the table?

**Response:** The R index cannot be calculated for P5, as all values of P5 refer to dry days with zero precipitation and no bias. This is not referred to anymore in the text, and P5 is not used in the evaluation. Hence, we followed the suggestion to remove P5 from the table.

Comment: L 405: Consider separating table 3 and table 4 results into two paragraphs.

**Response:** The discussion of these tables is now split into two paragraphs.

Comment: Figure 1: Panel f is missing.

**Response:** We thank the reviewer for noticing this. The figure has been updated

Comment: L 437: Why can't these percentiles be plotted, is it by choice? If this is the case, a reference in the text justifying the decision to limit the figure's range would be useful.

**Response:** This is indeed by choice. The RB_MB or RB_O values of these percentiles are (much) larger than 1, and thus plotting them wouldn't allow an easy comparison between either calibration and validation period or variables. This clarification has also been added to the text.

Comment: Figure 6 caption: R2D2 tag is missing next to (f).

**Response:** We thank the reviewer for noticing this, the caption has been updated accordingly.

Comment: L 528: It could be worth discussing that there is a seasonal component to the effect of ETP, could winter biases be more relevant for flood modelling than summer biases in ETP?

**Response:** This is indeed relevant, as causes and types of floods differ depending on the season. The difference between pluvial floods in summer and fluvial floods in winter, and how the PDM is primarily meant to capture the latter type of floods, is now discussed in the text. In addition, the text has been expanded to more extensively discuss the nonstationarity propagation of evaporation, as the biases are largest in spring, whereas the impact of soil moisture on floods is largest in winter.

Comment: Table 8: mQDM, Winter, Cal: value is 1%, should this be 100%?

**Response:** This should indeed be 100%, and has been updated accordingly.

Comment: L 627-660: This is a very long paragraph which can be split to ease reading (e.g., along sentences starting with First, Second, Third…).

**Response:** We thank the reviewer for providing this perspective. As guidance in the form of First, Second… was already present, the paragraph has now been split in several smaller paragraphs.

Comment: L 648: Related to the above, the sentence starting here is important and should be highlighted as an important result (perhaps by starting a new paragraph?)

**Response:** In addition to splitting the paragraph based on the four observations, the last part of the former paragraph, starting at line 648, has now been made a separate paragraph.